

# Evaluating the performance of the land surface model ORCHIDEE-CAN on water and energy flux estimation with a single- and a multi- layer energy budget scheme

Yiying Chen[1,*], James Ryder[1], Vladislav Bastrikov[1], Matthew J. McGrath[1], Kim Naudts[1,**], Juliane Otto[1,***], Catherine Ottlé[1], Philippe Peylin[1], Jan Polcher[2], Aude Valade[3], Andrew Black[4], Jan A. Elbers[5], Eddy Moors[5], Thomas Foken[6], Eva van Gorsel[7], Vanessa Haverd[7], Bernard Heinesch[8], Frank Tiedemann[9], Alexander Knohl[9], Samuli Launiainen[10], Denis Loustau[11], Jérôme Ogée[11], Timo Vesala[12,13], and Sebastiaan Luyssaert[1,****]

[1]Laboratoire des Sciences du Climat et de l'Environnement, LSCE/IPSL, CEA-CNRS-UVSQ, Université Paris-Saclay, F-91191 Gif-sur-Yvette, France
[2]Laboratoire de Météorologie Dynamique (LMD, CNRS), Ecole Polytechnique, Palaiseau, France
[3]Institut Pierre Simon Laplace, Place Jussieu 4, 75010 Paris, France
[4]Land and Food Systems, University of British Columbia, Vancouver, BC, Canada
[5]Alterra, Wageningen UR, Wageningen, the Netherlands
[6]Deptartment of Micrometeorology University of Bayreuth, Bayreuth Center of Ecology and Environmental Research, Bayreuth, Germany
[7]CSIRO, Marine and Atmospheric Research, Canberra, Australia
[8]Dept. Biosystem Engineering (BIOSE), University of Liege, Gembloux, Belgium
[9]Dept. Bioclimatology, Georg-August University of Göttingen, Büsgenweg Göttingen, Germany
[10]Natural Resources Institute Finland, Vantaa, Finland
[11]INRA UMR 1391 ISPA Centre de Bordeaux Aquitaine, Bordeaux, France
[12]Department of Physics, University of Helsinki, Helsinki, Finland
[13]Department of Forest Sciences, University of Helsinki, Helsinki, Finland
[*]now at: Graduate Institute of Hydrological and Oceanic Sciences, National Central University, Taiwan
[**]now at: Max Planck Institute for Meteorology, Hamburg, Germany
[***]now at: Climate Service Center Germany (GERICS), Helmholtz-Zentrum Geesthacht, Hamburg, Germany
[****]now at: Department of Ecological Sciences, VU University, Amsterdam, the Netherlands

*Correspondence to:* Yiying Chen (Yiying.Chen@lsce.ipsl.fr)



**Abstract.**

Canopy structure is one of the most important vegetation characteristics for land-atmosphere interactions, as it determines the energy and scalar exchanges between the land surface and the overlying air mass. In this study we evaluated the performance of a newly developed multi-layer energy budget in the land surface model ORCHIDEE-CAN (Organising Carbon and Hydrology

In Dynamic Ecosystems - CANopy), which simulates canopy structure and can be coupled to an atmospheric model using an implicit coupling procedure. We aim to provide a set of acceptable parameter values for a range of forest types. Top-canopy and sub-canopy flux observations from eight sites were collected in order to conduct this evaluation. The sites crossed climate zones from temperate to boreal and the vegetation types included deciduous, evergreen broad leaved and evergreen needle leaved forest with a maximum $LAI$ (all-sided) ranging from 3.5 to 7.0. The parametrization approach proposed in this study was

based on three selected physical processes − namely the diffusion, advection and turbulent mixing within the canopy. Short-term sub-canopy observations and long-term surface fluxes were used to calibrate the parameters in the sub-canopy radiation, turbulence and resistances modules with an automatic tuning process. The multi-layer model was found to capture the dynamics of sub-canopy turbulence, temperature and energy fluxes. The performance of the new multi-layer model was further compared against the existing single-layer model. Although, the multi-layer model simulation results showed little or no improvements

to both the nighttime energy balance and energy partitioning during winter compared with a single-layer model simulation, the increased model complexity does provide a more detailed description of the canopy micrometeorology of various forest types. The multi-layer model links to potential future environmental and ecological studies such as the assessment of in-canopy species vulnerability to climate change, the climate effects of disturbance intensities and frequencies, and the consequences of biogenic volatile organic compounds (BVOC) emissions from the terrestrial ecosystem.

**1   Introduction**

Today's Earth system models integrate ocean, ice sheet, atmosphere and land surface in order to provide a powerful tool to simulate the Earth's past, present and future climates (Drobinski et al., 2012). In such a model, the land surface sub-model provides the surface fluxes to the atmospheric sub-model, affects the dynamics of the planetary boundary-layer, and exerts a strong influence on the climate. The dynamics of the simulated surface fluxes rely on the land surface sub-model, that over

the past 40 years, has evolved from a simple bucket model approach towards sophisticated soil-vegetation-atmosphere-transfer (SVAT) schemes (Pitman, 2003; Stöckli and Vidale, 2005).

Although present day land surface models differ from each other in their formulation and details, their performance shows similar deficiencies. For example, imposing the same land cover changes to seven land surface models resulted in diverging climate effects. Among other factors, this divergence was due to the parametrization of albedo, and the representation of

evapotranspiration for different land cover types (Pitman et al., 2009). Difficulties in reproducing fluxes of sensible and latent heat for a wide range of vegetation types have been ascribed to the so-called 'big-leaf' approach (Bonan, 1996; Sellers et al., 1996; Dickinson et al., 1998; Jiménez et al., 2011) which treats the surface as a isothermal large leaf. Potentially, representing the vertical canopy structure in detail and simulating radiation partitioning and turbulent transport within the vegetation will



result in an improved determination of sensible and latent heat flux estimates (Baldocchi and Wilson, 2001; Ogée et al., 2003; Bonan et al., 2014). For example, several multi-layer SVAT schemes have been proposed and validated with site level observations (Ogée et al., 2003; Staudt et al., 2011; Haverd et al., 2012; Launiainen et al., 2015). These studies demonstrated that both top-canopy flux, within-canopy fluxes and micrometeorological profiles could be captured by means a sophisticated

parametrization scheme to describe the vegetation dynamics and the coupling between the atmosphere and the canopy.

Because the standard version of ORCHIDEE (Organising Carbon and Hydrology In Dynamic Ecosystems) makes use of a big-leaf approach (Ducoudré et al., 1993; Krinner et al., 2005), improved model capacity and performance were aimed for by implementation of a multi-layer energy budget scheme (Ryder et al., 2016) that was integrated with vertically discrete reflectivity, photosynthesis, stomatal resistance and carbon allocation schemes. This new design resulted in a new version of

ORCHIDEE named ORCHIDEE-CAN (ORCHIDEE-CANopy, revision 2290) (Naudts et al., 2015). Despite its code including a multi-layer energy budget scheme (Ryder et al., 2016), ORCHIDEE-CAN is currently applied using a single-layer energy budget, due to a lack of validated parameters for the multi-layer energy budget scheme.

In this study, we compiled a set of within-canopy and above-canopy measurements of energy, water and $CO_2$ fluxes and used these data to parametrize and validate the new multi-layer energy budget scheme for a range of forest types. An adequate

parametrization approach will be also presented for the global scale land surface model ORCHIDEE-CAN (revision 2754) that was applied in this study. Furthermore, model performance of the new multi-layer parametrization was compared against the existing single-layer model.

## 2 Methodology

### 2.1 Multi-layer energy budget scheme

The multi-layer energy budget scheme used in this study was developed for global land surface models (Ryder et al., 2016) and the calculations differ from the more common big-leaf energy budget scheme in three aspects: The new scheme calculates: (a) a within-canopy longwave and shortwave radiation based on a vertical leaf area index ($LAI$; $\mathrm{m}^2\,\mathrm{m}^{-2}$) profile, (b) a within-canopy and below-canopy wind profile based on the vertical $LAI$ profile and (c) the dependency of stomatal resistance and aerodynamic resistance based on the microclimatological conditions along the $LAI$ profile. All symbols are explained in Table

1. In the following paragraphs these calculations are further described.

(a) The multi-layer energy budget scheme makes use of the longwave radiation transfer scheme proposed by Gao et al. (1989) and Gu et al. (1999). The scheme simulates longwave radiation transport, as well as scattering and absorption, along a vertically layered leaf area distribution. The simulated longwave radiation within a layer depends on the emitted longwave radiation by all of its neighbouring layers. The shortwave radiation transfer scheme, developed by Pinty et al.

(2006), was applied to the albedo calculation. The scheme computes the absorption, transmission, and reflection of incoming radiation by vegetation canopies, which depends on the solar zenith angle, the type of illumination (direct or diffuse), the vegetation type, and the vegetation structure. This scheme considers shortwave radiation both from visible





and near infrared bands and was originally developed for single-layer canopies, but has since been extended for use with layered canopies (McGrath et al.).

(b) The wind profile and the vertical eddy diffusivity ($k$; $\mathrm{m^2\,s^{-1}}$) are calculated using the one-dimensional second-order closure model of Massman and Weil (1999), which makes use of the $LAI$ profile of the stand. It calculates wind profile and vertical eddy diffusivity based on Lagrangian theory.

(c) The aerodynamic resistance ($R_b$; $\mathrm{s\,m^{-1}}$) is calculated based upon the leaf boundary-layer resistance, which is estimated according to Baldocchi (1988). The stomatal resistance ($R_s$; $\mathrm{s\,m^{-1}}$) is calculated using a Farquhar-von Caemmerer-Berry-type C3 (Farquhar et al., 1980) and Collatz-type C4 photosynthesis model (Collatz et al., 1992) which simultaneously solves carbon assimilation and stomatal conductance at the leaf level but excludes mesophyll conductance calculation. ORCHIDEE-CAN uses an analytical approach as described by Yin and Struik (2009) to calculate layered stomatal resistances which depend on the ambient air temperature, humidity, within-canopy $CO_2$ concentration, vegetation-specific maximum carboxylation rate, and water supply from the roots to the stomata.

Readers are referred to Ryder et al. (2016) for a comprehensive description of the multi-layer energy budget, its assumptions, mathematical details and a proof of concept. Note that in ORCHIDEE-CAN $LAI$ is calculated from a prognostic leaf mass by making use of a vegetation-specific specific leaf area ($SLA$; $\mathrm{m^2\,g^{-1}}$). The calculation of the vertical and horizontal distribution of the leaf mass, and thus the vegetation canopy depends on plant phenology, intra-stand competition, forest management, and allometric relationships, and is detailed in Naudts et al. (2015).

## 2.2 Observational data

For this study forest sites were retained if the following data were available: (a) short but intensive campaigns making flux and profile measurements within and/or below the tree canopy and, (b) multi-year monitoring of top-canopy fluxes. Through numerous regional projects such as CARBOEUROPE, AMERIFLUX, Fluxnet Canada, OZFLUX, ICOS and NEON, and efforts such as FLUXNET (Baldocchi and Wilson, 2001), multiple year-long time series are now commonly available especially for the temperate and boreal zones in Europe, Japan, Australia and North America. Site selection was thus mostly limited by the availability of within-canopy and below-canopy measurements.

Eight flux observation sites (Table 2) met the aforementioned criteria, and represented various climates from the Mediterranean to the boreal zone and different vegetation types including broad-leaved summer green, broad-leaved evergreen and needle-leaved evergreen. Data were thus missing from needle-leaved summer green vegetation such as Larch (*Larix sp.*) and tropical vegetation, so it was not possible to cover all of the forest types that are considered in ORCHIDEE-CAN.

The short intensive campaigns making measurements within-canopy and below-canopy usually extended for periods ranging from several days to a few weeks (Period I; Table 3). During intensive campaigns, vertical profile measurements of wind speed, temperature and atmospheric humidity were typically conducted. Such measurements were sometimes complemented with profile measurements of sensible and latent heat fluxes, as well as sub-canopy radiation measurements (Period II and III; Table 3). Furthermore, our parametrization and validation set-up required that top-canopy observations had to be available for periods



exceeding one year (Period IV; Table 3). A typical long-term set-up measured sensible and latent heat fluxes, longwave and shortwave incoming radiation, wind speed, atmospheric temperature and humidity.

Parametrization and validation utilises the ORCHIDEE-CAN model simulations, and so climate forcing data were required to drive the simulations. Site-level weather observation, i.e., shortwave incoming radiation, longwave incoming radiation, two dimensional wind speed, precipitation, snow, near-surface air pressure and specific humidity were reformatted and gap-filled using the method proposed by Vuichard and Papale (2015). Weather observations are an integral part of both intensive campaigns and multi-year top-canopy flux monitoring. Hence, within a measurement site, flux, profile, and weather data were usually available at the same temporal resolution and over the same time periods.

Finally, the forcing files were completed with the observed vertical $LAI$ profiles. However, the temporal resolution of $LAI$ was much lower than the resolution of the meteorological variables. When the total $LAI$ was measured at a higher time resolution than its vertical profile, the observed total $LAI$ was vertically distributed according to the observed relative vertical $LAI$ distribution. Model parametrization (section 2.3) and model experiments that aimed at testing the performance of only the multi-layer energy budget (section 2.5) made use of the observed $LAI$ profiles. For the remaining two model experiments, (section 2.5) ORCHIDEE-CAN calculated the vertical $LAI$ profiles following the carbon allocation and carbon turnover schemes, as described in Naudts et al. (2015).

### 2.3 Model parametrization

At the start of this study the multi-layer energy budget did not yet have a working set of parameters for ORCHIDEE-CAN. Therefore, we refrained from performing a sensitivity analysis prior to optimizing the model parameters (Kuppel et al., 2014; MacBean et al., 2015) but instead selected three processes, described by a total of 10 parameters for optimization. The selected processes were related to the physical processes within the canopy, i.e., diffusion, advection and turbulent mixing.

#### 2.3.1 Effective drag coefficient $C_{Deff}$ (unitless)

A one-dimensional second-order closure wind profile model (Massman and Weil, 1999) was used to estimate the vertical within-canopy wind profile. This approach requires an effective drag coefficient, which relates to the vertically discretised estimate of the canopy drag coefficient ($C_{D,i}$; unitless ) and the momentum shielding factor ($P_{m,i}$; unitless) as follows:

$$C_{Deff,i} = C_{D,i}/P_{m,i} \tag{1}$$

Both the within-canopy drag and the momentum shielding were parametrized as an effective drag coefficient using a function of cumulative leaf area index ($LAI_{cum}$; $\mathrm{m^2\,m^{-2}}$) from the top canopy layer to the bottom layer, which was modified from the original function (Wohlfahrt and Cernusca, 2002) as below:

$$C_{Deff,i} = a_1^{-LAI_{cum,i}/a_2} + a_3^{-LAI_{cum,i}/a_4} + a_5 \tag{2}$$





where the subscript $i$ denotes the index of layering from the bottom layer ($i = 1$) to the top-canopy layer ($i = n$). $a_1$ to $a_5$ are tuning coefficients (unitless). The default parameter values for $a_1$ to $a_5$ are presented in Table 4.

### 2.3.2 Eddy diffusivity for vertical energy and water transport $k$ (m$^2$ s$^{-1}$)

After the vertical wind profile was derived from the one-dimensional second-order closure wind profile model, the friction velocity ($u_*$, m s$^{-1}$), the vertical wind velocity variance ($\sigma_w$; m s$^{-1}$) and Lagrangian time scale ($T_L$; s) were calculated following the approach by Raupach (1989). In this approach the vertical eddy diffusivity is a function of $\sigma_w$ and $T_L$. Subsequently, the vertical eddy diffusivity down the air column to the forest floor was calculated as follows:

$$k_i = \sigma_{w,i}^2 T_{L,i} \tag{3}$$

The relationship between atmospheric conditions and within-canopy transport is well documented (Raupach et al., 1996), but remains poorly understood. One compromise to accommodate this lack of detail is to apply a different scaling for $k_i$, according to the time of the day. Here we build on a similar approach but, rather than using time of the day, we used the calculated friction velocity ($u_* = u(h_c) * (0.32 - 0.264e^{-15.1\zeta(h_c)}$ where $\zeta$ is the cumulative function of $D_{eff}$, and $h_c$ is the canopy height.) to account for the observed differences in vertical transport within the canopy between daytime and nighttime by applying a weighting factor ($W_{nf}$; unitless). Therefore the modified diffusivity for level $i$ ($k_i^*$ ; m$^2$ s$^{-1}$) was defined as:

$$k_i^* = W_{nf} k_i \tag{4}$$

where $W_{nf}$ was calculated as:

$$W_{nf} = \frac{1}{1 + e^{(-a_6(u_* - a_7))}} \tag{5}$$

This function has a sigmoidal shape, where $a_6$ is the ceiling factor of the slope, and $a_7$ is the critical friction velocity at the inflection point of the sigmoid function (Fig. 1A). Consequently, atmospheric diffusivity is reduced if $u_*$ is low, which represents stable atmospheric conditions. Under turbulent atmospheric conditions, which are represented by a high $u_*$, $W_{nf}$ is close to one and the simulated diffusivity will closely follow the relationship proposed by Raupach (1991). Within-canopy transport is far-field dominated and the eddy diffusivity was calculated as a function of friction velocity, standard deviation of vertical wind speed, observation height, and canopy height Haverd et al. (2012)):

$$k_i = 0.66\sigma_w^2 \left(\frac{h_c}{u_*}\right)\left(\frac{1 - e^{-4.86z/h_c}}{1 - e^{-4.86}}\right) \tag{6}$$

The default parameter values for $a_6$ and $a_7$ are presented in Table 4. As an alternative to using $u_*$, it has been proposed to use a mixing length scale to classify flow regimes in order to give a better description of the coupling process below and above the





forest canopy (Thomas and Foken, 2007; Staudt et al., 2011; Foken et al., 2012). The numerical scheme of this approach relies on iterations. Since ORCHIDEE-CAN is designed to be coupled to regional or global atmospheric models, its numerics has been designed to avoid iterations in order to run efficiently.

### 2.3.3 Conductance for the soil-atmosphere interface $k_{surf}$ (m s$^{-1}$)

In Mediterranean, temperate, and boreal forests the characteristics of the interface between the soil and the atmosphere will change with the seasons following the under-story phenology. In winter, when the under-story is senescent, the characteristics in terms of the evapotranspiration at the interface will closely resemble the evapotranspiration of a bare soil. In summer, however, an under-story will be present and its density relates to the gap fraction of the over-story canopy. Hence, the summertime evapotranspiration of the interface will be more similar to the evapotranspiration of a vegetation canopy. Therefore, we intro-

duced $\beta_0$ (unitless) as a weighting function ranging from zero to unity, in order to scale the surface conductivity as a function of under-story phenology. Under-story phenology was described as a function of the over-story canopy coverage $(1 - f_{Pgap})$ and the mean air temperature during the previous 21 days $(\overline{T}_a)$:

$$\beta_0 = \begin{cases} \frac{a_{10}}{(1+\frac{298.15-\overline{T}_a}{15})+e^{(-a_8((1-f_{Pgap})-a_9))}}, \text{ when } \overline{T}_a \leq 298.15 \\ \frac{a_{10}}{1+e^{(-a_8((1-f_{Pgap})-a_9))}}, \text{ when } \overline{T}_a > 298.15 \end{cases} \qquad (7)$$

where $a_8$ is a factor that constrains the slope of the function and $a_9$ is a threshold for the vegetation cover. $a_{10}$ is a linear

weighting factor. $f_{Pgap}$ is calculated in ORCHIDEE-CAN and describes the over-story gap probability, which is a function of the canopy structure of the vegetation and the solar zenith angle and is calculated in ORCHIDEE-CAN. The weighting factor $W_{sf}$ for the soil-atmosphere interface is described as the conditional function of canopy cover fraction $(1$-$f_{Pgap})$ $W_{sf} = \beta_0$ when $(1 - f_{Pgap} > a_9$; and $W_{sf} = 1 - \beta_0$ when $(1 - f_{Pap}) \leq a_9$ (see Fig. 1B). For the lowest layer in the air column, i.e., the layer adjacent to the surface, the surface conductance is then calculated as:

$$k_{surf} = (W_{sf}\beta_3 + (1 - W_{sf})\beta_4)(u_1 C_{Deff,1}) \qquad (8)$$

where $\beta_3$ and $\beta_4$ are coefficients respectively describing the fraction of the potential plant transpiration and soil evaporation that are realized . The definition of these coefficients and the numerical approaches are presented in Ryder et al. (2016) and Dufresne and Ghattas (2009). $u_1$ is the wind speed at the lowest canopy layer thus close to the forest floor and is derived from the one-dimensional second-order closure model. $C_{Deff}$ is the effective drag coefficient calculated according to Eq.1. The

default parameter values of $a_8$, $a_9$, $a_{10}$ and $W_{sf}$ are presented in Table 4.





### 2.3.4 Boundary-layer resistance of the leaf surface $R_b$ $(\mathrm{s\,m^{-1}})$

The boundary-layer resistance of the leaf surface $R_{b,i}$ is described according to the expression from Baldocchi (1988):

$$R_b = \begin{cases} W_{br}(\frac{d_l}{D_{h,air}Nu}), \text{ for sensible heat} \\ W_{br}(\frac{d_l}{D_{h,H2O}Sh}), \text{ for latent heat} \end{cases} \tag{9}$$

where $W_{br}$ accounts for the fact that the leaf length of the species under study differs from the characteristic leaf length
(unitless), $d_l$ is the characteristic leaf length (0.001 m was used as the default value), $D_{h,air}$ is the heat diffusivity of still air
$(\mathrm{m^2\,s^{-1}})$, $D_{h,H2O}$ is the heat diffusivity of water vapor $(\mathrm{m^2\,s^{-1}})$, $Sh$ is the Sherwood number (unitless), and $Nu$ is the Nusselt
number (unitless). The Sherwood number was calculated as $Sh = 0.66\,Re^{0.5}\,Sc^{0.33}$ for laminar flow and $Sh = 0.03\,Re^{0.8}\,Sc^{0.33}$
for turbulent flow, where $Sc$ is Schmidt number (0.63 for water vapor; unitless). The transition from laminar to turbulent
flow takes place in the model when the Reynolds number exceeds a value of 8000. The Nusselt number was calculated as
$Nu = 0.66\,Re\,Pr^{0.33}$ , where $Pr$ is Prandtl number (0.7 for air; unitless)(Grace, 1978), and $Re$ is the Reynolds number (unit-
less) which was calculated as:

$$Re = \frac{d_l\,u_i}{\mu} \tag{10}$$

where $u_i$ is the horizontal velocity at level $i$ $(\mathrm{m\,s^{-1}})$ and $\mu$ is the kinematic viscosity of air and was set to 0.0015 $(\mathrm{m^2\,s^{-1}})$
(Garratt, 1992). The default parameter value for $W_{br}$ is provided in Table 4.

### 2.3.5 Stomatal resistance $R_s$ $(\mathrm{s\,m^{-1}})$

The stomatal resistance of the leaves was calculated for each canopy layer based on the parameters within the layer under
consideration. Two stomatal resistances were calculated with the adjusted assimilation rate: (a) the stomatal resistance assuming
unlimited water availability and (b) the stomatal resistance that exactly satisfies the amount of water the plant can transport
from its roots to its stomata. The largest of the two resistances and the concurrent $CO_2$ assimilation and transpiration rate were
then used in the remainder of the model calculations. This approach is detailed in Naudts et al. (2015) and the numerical scheme
for its multi-layer implementation is given in Ryder et al. (2016).ORCHIDEE-CAN scales stomatal resistance to account for
the part of the canopy that is coupled to the atmosphere and thus contributes to the latent heat flux. In this study, this weighting
was formalized through a linear parameter $W_{sr}$:

$$R_{s,i} = W_{sr}(\frac{1}{(g_0 + (\frac{A_i h_s}{C_s}))LAI_i}) \tag{11}$$

where $g_0$ is the residual stomatal conductance if the solar irradiance approaches zero, $C_s$ is the concentration of $CO_2$ at the leaf
surface and $h_s$ is the relative humidity at leaf surface. $A$ is the $CO_2$ assimilation rate which is solved analytically following



(Yin and Struik, 2009). In Eq. 11 the relative humidity used is the top canopy forcing instead of a layered relative humidity in order to avoid an iterative process. The default parameter value for $W_{sr}$ is presented in Table 4.

## 2.4 Model optimization

### 2.4.1 Optimization procedure

Parametrizing the scaling coefficients and weighting factors enabled us to simultaneously improve the match between the simulated and observed sub-canopy micrometeorology, including temperature and specific humidity when available, and between the simulated and observed top-canopy heat fluxes ($LE$ and $H$). Within-canopy fluxes were also simulated but are not usually measured. The parametrization made use of an in-house optimization package called ORCHIDAS (ORCHIDEE Data Assimilation Systems;http://orchidas.lsce.ipsl.fr/). ORCHIDAS provides a range of numerical approaches for assimilating multiple

data streams in ORCHIDEE.

We used the maximum gradient approach to tune the parameters $a_3$ to $a_{10}$, $W_{br}$, and $W_{sr}$ for each study site independently. Over the course of several iterations, the optimization approach minimized the mismatch between the model output and the observations, using a gradient based algorithm called L-BFGS-B (Limited-memory Broyden-Fletcher-Goldfarb-Shanno algorithm with Bound constraints), which provides the possibility to prescribe boundaries for each parameter (Byrd et al., 1995).

The range assigned to each parameter is reported in Table 4. Furthermore, this approach allowed for measurement uncertainties in the eddy covariance $LE$ measurement by reducing its weight in the cost function from 1.0 to 0.66. This value of 0.66 was set based on the outcome of a paired tower-experiment to estimate the random errors of the eddy covariance measurements (Richardson et al., 2006). For the optimisation the $LAI$ in ORCHIDEE-CAN was set to match the observed vertical $LAI$ profile.

A three-step optimization procedure was carried out in this study. Firstly, the within-canopy and below-canopy observations from the short-term intensive measurement campaigns (Period I in Table 3) were used to optimise $a_3$ to $a_7$, $W_{br}$ and $W_{sr}$. During this step, the parameters for the soil-atmosphere interface ($k_{surf}$, i.e. $a_8$ to $a_{10}$ and $W_{sf}$) were set to their default values. Due to the fact that these campaigns took place during summer, parameters related to the within-canopy effective drag profiles, eddy diffusivity, boundary layer resistance and stomatal resistance ($C_{Deff}$; $k$; $R_b$; $R_s$) were biased towards the

summer. Secondly, the seasonal dynamics of $k_{surf}$ was parametrized by trying to improve the correspondence between the simulated and observed top-canopy fluxes over one year (Period II in Table 4). In this step $a_3$ to $a_7$, $W_{br}$ and $W_{sr}$ were set to the values obtained from the first step of the optimization and $a_8$ to $a_{10}$ and $W_{sf}$ were tuned. Finally, performance of the calibrated model was evaluated based on a second single year of top-canopy observations (Period III in Table 3).

Although the spin-up was stopped on June 30th (Table S1 in the Supplementary Information) and all simulations thus used

the June 30th soil water content as their initial condition, this approach does not guarantee that this typical summer soil water content matches the soil water content in the year of the intensive measurement campaign. The effect of this possible mismatch was quantified by running a sensitivity analysis in which the whole parametrization approach, which was repeated for seven different initial soil water contents − varied from -30% to 30% in increments of 10% of the June 30th value.



## 2.5 Attribution of changes in model performance

The multi-layer energy budget scheme (Ryder et al., 2016) that was parametrized and tested in this study required realistic spatially and temporally soil water content and a value for the ground heat flux from surface level as initial conditions. This need was satisfied by implementing this scheme within the newly enhanced land surface model ORCHIDEE-CAN (Naudts et al., 2015). Integration of the multi-layer energy budget in ORCHIDEE-CAN, however, complicated the design of the validation study as it was now necessary to separate, as much as possible, the performance of the multi-layer energy budget scheme from the performance of the rest of the model. To this aim, four experiments were designed in order to better understand the performance of the new scheme (Table S1 in the Supplementary Information).

**Experiment 1 (EXP1): Single-layer scheme with a prescribed canopy**

The first experiment was run at the site-level and made use of the default single-layer energy budget scheme. The energy budget scheme was driven by the observed climate forcing and the observed total $LAI$ (Table 2). In this experiment, the vertical $LAI$ profile was only used for the photosynthesis module in ORCHIDEE-CAN. Note that vertical $LAI$ profiles cannot be used by the single-layer scheme and the results are therefore limited to the top-canopy fluxes. This experiment was used as the reference simulation to document the performance of the single-layer approach.

**Experiment 2 (EXP2): Single-layer scheme with a simulated canopy**

The second experiment was identical to the first experiment except that the $LAI$ was now simulated by ORCHIDEE-CAN, rather than using the observed $LAI$. Given that these experiments make use of observed climate drivers and $LAI$, changes in model performance between experiment 1 and 2 are derived by the introduction of a dynamic and prognostic vertical $LAI$ profile. A large decrease in performance between experiments 1 and 2 would suggest that ORCHIDEE-CAN does a poor job in simulating the vertical $LAI$ profile.

**Experiment 3 (EXP3): Multi-layer scheme with a prescribed canopy**

Experiment 3 differs from EXP1 through the use of the multi-layer energy budget scheme, rather than the single-layer scheme. As a consequence, the observed vertical $LAI$ profiles rather than the observed total $LAI$, is now applied to drive the simulations with a multi-layer energy budget. This experiment was used for quantifying the change in performance when switching from the single-layer to the multi-layer approach. Although these simulations calculate the turbulent fluxes for each canopy level, the change in performance was based on a comparison of experiment 1 and 3, and as such the analysis had to be limited to the top-canopy fluxes, as within-canopy fluxes cannot be calculated by the single-layer approach used in the first experiment. A large decrease in performance between experiment 1 and 3, would suggest that the multi-layer energy budget in ORCHIDEE-CAN does not help to better simulate the top-canopy fluxes.

**Experiment 4 (EXP4): Multi-layer scheme with a simulated canopy**

In Experiment 4 the vertical $LAI$ profile was calculated by ORCHIDEE-CAN. Thus, this experiment made use of the full functionality of ORCHIDEE-CAN and the multi-layer energy budget. As such, albedo, photosynthesis and the energy budget calculations were fully consistent. Comparing the performance of experiments 2 and 4 quantifies the



actual change in performance for a prognostic $LAI$ profile and its interactions in ORCHIDEE-CAN. A large decrease in performance between experiment 2 and 4 would therefore suggest that the multi-layer energy budget in ORCHIDEE-CAN does not help to better simulate the top-canopy fluxes. Furthermore, a large decrease in performance between experiments 3 and 4 would indicate that ORCHIDEE-CAN does a poor job in simulating the vertical $LAI$ profile.

5    All four experiments were started from 20 years spin-up simulations, which were driven by CRU-NCEP climate re-analysis from 1991 to 2010 with a spatial resolution of $0.5°$x $0.5°$(Maignan et al., 2011) at selected study sites. These spin-up simulations allow the model to build-up a realistic soil water pool at the start of each simulation. A ten-layer $LAI$ profile was applied for each site - the number of layers chosen follows the approach from a previous study (Ryder et al., 2016). If the vertical $LAI$ profile was prescribed, the total $LAI$ was re-scaled within these ten layers to follow the observed vertical $LAI$ profile at each

10   site (Fig. 2). If the vertical $LAI$ profile was not imposed, the $LAI$ generated for the albedo calculation (McGrath et al.) was used instead. Note that contrary to previous versions of ORCHIDEE, ORCHIDEE-CAN no longer applies a constraint on the maximum $LAI$. In ORCHIDEE-CAN, the total $LAI$ is the outcome of carbon allocation to the canopy through a pipe-model and carbon removal from the canopy through leaf turnover (Naudts et al., 2015).

### 2.6   Model performance

15   The change in model performance due to the use of the multi-layer rather than the single-layer scheme for a prescribed $LAI$ profile (EXP1 vs. EXP3), and a simulated $LAI$ profile (EXP2 vs. EXP4), were quantified by comparing the Taylor skill score ($S_T$) (Taylor, 2001).

$S_T$ was calculated for the eight observational sites for the top-canopy fluxes of all four experiments making use of the simulated and observed half-hourly fluxes. The Taylor skill score was calculated as follows:

$$S_T = \frac{4(1+R)}{(\hat{\sigma}_f + 1/\hat{\sigma}_f)^2(1+R_0)} \tag{12}$$

where, $R$ is the correlation coefficient between the simulation and the observation, $R_0$ is the maximum correlation coefficient and $\hat{\sigma}_f$ is the ratio of the variance of the simulations to the variance of observations ($\hat{\sigma}_f = \sigma/\sigma_r$). Here, we set $R_0$ to 1.0 for the maximum correlation between observation and model simulation. A value of 1.0 of $S_T$ indicates that model simulations perfectly matches the observations, values lower than 0.5 imply that the model has poor predictive ability.

## 3   Results

### 3.1   Model parametrization

Using the default parameter set (i.e., $a_1$ to $a_5$) resulted in an underestimation of the wind speed in the lower canopy level at all study sites. Optimized parameters could be roughly grouped according to canopy structure (see Table S1 in the Supplementary Information). For forest sites with a dense canopy (see the second low of Fig. S1 in the Supplementary Information), the param-





eters had to be adjusted to simulate a low wind speed in the lower canopy. For forest sites with a sparse canopy, the parameters had to be adjusted to simulate relatively high wind speeds at the bottom of the canopy. At these sites, flux observations showed a substantial contribution from the forest floor to the sensible and latent heat fluxes at the top of the canopy. The average model error of wind profile estimation, in terms of root mean square error (RMSE), was reduced from $0.62 \, \mathrm{m \, s^{-1}}$ to $0.42 \, \mathrm{m \, s^{-1}}$ after adjusting the parameters (see Table S3 in the Supplementary Information). Tuning the conductance of the soil-atmosphere interface (i.e., $a_8$ to $a_{10}$), rather than tuning the stomatal conductance and leaf boundary-layer resistances, enabled a closer match between the simulations and observations (Figs. S2 and S3 in the Supplementary Information).

At sites with dense canopies, however, tuning the weightings of stomatal resistance and weighting the boundary layer resistance improved the match between the simulated and observed inner-canopy and top-canopy fluxes of sensible and latent heat (Figs. S2 and S3 in the Supplementary Information). The model errors of heat and water fluxes estimations were reduced substantially from $91.2 \, \mathrm{W \, m^{-2}}$ to $46.1 \, \mathrm{W \, m^{-2}}$ for $LE$ and $123.2 \, \mathrm{W \, m^{-2}}$ to $50.3 \, \mathrm{W \, m^{-2}}$ for $H$, respectively (also see the Table S3 in the Supplementary Information).

At sites with sparse canopies, the net radiation at the forest floor was substantial, i.e., ranging nearly from $200 \, \mathrm{W \, m^{-2}}$ to $450 \, \mathrm{W \, m^{-2}}$ (Fig. S4 in the Supplementary Information). Correctly simulating radiation transfer strongly contributed to correctly simulating the within-canopy flux profiles and top-canopy latent and sensible heat fluxes. Nevertheless, radiation transfer was not re-parametrized in this study and, hence, the model errors of net radiation estimation depended solely on the tree species. In sparse canopies, a positive air temperature gradient with higher temperatures at the forest floor compared to the top-canopy was also presented (Fig. S5 in the Supplementary Information). Using default parameter values for all factors resulted in a good simulation of the air temperature gradient for all eight sites. However, optimizing the parameters (i.e., $a_3$ to $a_{10}$, $W_{br}$ and $W_{sr}$) had a large impact on the absolute values of the vertical profile in leaf temperature (Fig. S6 in the Supplementary Information). Leaf temperature was not measured at any of the sites. Therefore, it remains to be assessed whether the model can concurrently reproduce observed energy fluxes and soil water contents.

At one site with an open canopy (FR-LBr) the effect of the initial soil water content on the optimized parameter estimates was tested. Both the stomatal resistance and the boundary resistance weighting factors ($W_{sr}$ and $W_{br}$) were found to be very sensitive to the optimisation procedure with changes in their values exceeding 5% (Fig. S7 in the Supplementary Information). After parameter adjustment the sensitivity to initial soil water content was 5% less than that using the originally optimized values. Changes in parameters $a_6$ and $a_7$, which tuned the eddy diffusivity, were largely unaffected by the initial conditions. Soil water content measurements would thus have helped to improve the parametrization, especially for the stomatal and leaf boundary-layer resistances.

## 3.2 Performance of the single-layer scheme

Model performance of the single-layer model was evaluated making use of EXP1. Overall model performance for sparse canopies (Fig. 3A) was slightly higher and thus better than model performance at the dense forest sites (Fig. 3B). Moreover, model performance at the forests with sparse canopies showed less variability within a year than model performance at sites with a dense canopy.



At the sparse canopy sites, both the intra-annual and diurnal variation in net radiation $R_n$ was well simulated, displaying $S_T$ scores continuously over 0.9 (Figs. 3B and 3D). For dense canopies, the $S_T$ score of $R_n$ dropped to 0.9 in winter, which might be attributed to an incorrect estimation of $R_n$ during nighttime (Fig. 3C).

In general, the $S_T$ for the single-layer or big-leaf model for the sensible heat flux was higher than for the latent heat flux
both at the annual and daily resolution. The $S_T$ dropped below 0.5 for latent heat flux and 0.8 for sensible heat flux (Fig. 3A) from November to January (or May to July at Au-Tum), indicating that the single-layer model incorrectly partitioned energy during the cold season. During these months nights are long and the inability of the model to simulate nighttime fluxes (Fig. 3C) may well be the cause of the observed model deficiencies during the winter months.

### 3.3 Performance of the multi-layer scheme

Model performance of the multi-layer model was evaluated making use of EXP3. By introducing the multi-layer energy budget scheme, model performance for sparse and dense canopies became more comparable (Figs. 4A and 4B) due to small improvements in the $S_T$ for simulation of dense canopies and small losses in the skill to simulate the energy budget of sparse canopies. Improved simulations of nighttime fluxes under dense canopies (Fig. 4C) were reflected in the improved partitioning of energy fluxes during wintertime (compare Fig. 3A and Fig. 4A). The multi-layer energy budget model lost some skills compared
to the single-layer model in the simulation of the latent heat flux from sparse canopies between September and December. The discrepancy is mainly due to the loss of model performance for one deciduous forest sites (Fig. S8 in the Supplementary Information).

Overall, the introduction of the multi-layer energy budget and its integration in ORCHIDEE-CAN resulted in a small decrease in model skill (Fig. 5; Table S4 in the Supplementary Information). When moving from the single-layer scheme with a
prescribed $LAI$ (EXP1) to the multi-layer scheme with a simulated $LAI$ profile (EXP4), the model skill decreased for $R_n$, $H$, and $LE$ but increased for $G$. Despite this improvement, the overall model performance on the ground heat flux estimation at all eight forest sites was still very low < 0.5 (Figs. 5B-D; Table S4 in the Supplementary Information). The low performance may be due to either deficiencies in the model or inability of point measurements to represent the large variation in ground heat fluxes underneath a canopy or the errors made in estimating the rate of heat storage change in the layer of soil between
the soil heat flux plates and the soil surface (Mayocchi and Bristow, 1995; Kustas et al., 2000). However, the small loss (all fluxes except $G$) or gain (only for $G$) in model skill from introducing the multi-layer scheme can be strengthened (i.e., $LE$) or compensated for ($R_n$, $H$ and $G$) by the small gain in model skill from the introduction of a prognostic vertical $LAI$ profile.

## 4 Discussion

### 4.1 Single-layer v.s. multi-layer energy budget

Three major deficiencies of the single-layer energy budget scheme have been identified: (1) poor model performance in the net radiation estimation during nighttime in dense canopy forests; (2) incorrect energy partitioning during winter seasons at



dense forest sites and; (3) incorrect simulation of soil heat flux for all forest sites. These site-level findings are consistent with previous large-scale validation work (Pitman et al., 2009; Jiménez et al., 2011; de Noblet-Ducoudré et al., 2012) which applied the single-layer energy budget to simulate land surface fluxes dynamically and demonstrated that this approach has difficulties to in the reproduction of surface energy fluxes.

In this study, we tried to overcome these difficulties by implementing a multi-layer energy budget scheme. The multi-layer energy and water calculations make use of a vertically resolved radiation transfer scheme for shortwave and longwave radiation (replacing prescribed shortwave reflection values), a within-canopy wind velocity profile (replacing empirical formulations for roughness length), a vertical prognostic $LAI$ profile (replacing a prescribed $LAI$ value), within-canopy leaf boundary-layer resistance profiles for energy and water transport, a within-canopy stomatal resistance profile, a vertical discrete eddy diffusivity
profile and a soil-atmosphere layer conductivity.

This approach resulted in small improvements in simulating energy partitioning during nighttime for dense canopies, small losses in model performance in terms of energy partitioning for sparse canopies and year round gains in model performance for simulation of the ground heat flux. As such, the multi-layer energy and water vapor flux scheme did not solve the long-standing issues related to simulating nighttime energy partitioning (Jordan and Smith, 1994; Prihodko et al., 2008; Wild, 2009; He et al.,
2011) but it succeeded in obtaining a similar model performance while much of the empiricism of the big-leaf approach was replaced by a more realistic process description. A more realistic model description opens new avenues of research (see section 4.3).

## 4.2    Parametrization approach

Parametrization of the new scheme and its underlying processes revealed strengths and weaknesses of the model as well as
avenues for future experimental work.

(1) Within-canopy drag

For the inner-canopy drag parametrization, we modified an approach (Eq. 2) that has previously only been tested and validated at grassland sites (Wohlfahrt and Cernusca, 2002). In that study, $LAI$ was treated as equal to the plant area index ($PAI$), which is a separate measure that accounts not only for leaves but also for other vegetation material such as
stems and seedheads. In forests, however, the difference between $LAI$ and $PAI$ is made up by the branches and trunks and becomes especially important in winter in deciduous stands as canopy drag still exists. As a first parametrization this simplification allowed a better comparison with the observations and with the single-layer model. We applied a formulation that makes use of $LAI$ and, by doing so, some model errors might have been introduced, especially for the deciduous forest sites. ORCHIDEE-CAN now simulates both $LAI$ and $PAI$ and so this enhanced approach could be
adopted. Results confirmed that substituting $PAI$ by $LAI$ is acceptable during the leaf-on seasons (see Fig. S8 in the Supplementary Information).

Alternative approaches have been proposed by Cescatti and Marcolla (2004). For example, the inner-canopy drag could also be modelled as the function of the percentage of horizontal gaps in the forest canopy − a canopy characteristic that





is presently simulated in ORCHIDEE-CAN. Measurement sites such as DE-Bay or AU-Tum have detailed wind and vertical $LAI$ profile observations and could thus be used in a pilot study for developing a suitable parametrization approach linking inner-canopy drag and shielding to the canopy gaps. Such a development would also meet the requirements for calculating drag and shielding following small scale mortality from forest management, fires, wind damages and pests.

(2)  Within-canopy transport

In this study, within-canopy transport was parametrized by K-theory. A one-dimensional second-order closure model was applied to derive the within-canopy turbulence statistics, based both on the $LAI$ profile and the canopy height. This approach has been reported to produce a reasonable approximation of above-canopy fluxes estimation, even if the within-canopy temperature and humidity gradients are not always well captured (Raupach, 1989). As previous studies

have demonstrated, incorrect estimation on gradients may be accommodated to some extent by introducing a scaling factor (Eq. 5) to constrain the within-canopy transport (Makar et al., 1999; Wolfe et al., 2011; Ryder et al., 2016). Alternatively, such a scaling factor might vary in terms of the form of the canopy structure or openness though the determination of the factor has yet to be adequately described due to a restricted range of measurements (McNaughton and Van Den Hurk, 1995; Stroud et al., 2005).

At sparse forest sites, the temperature measurements showed a general positive gradient during the daytime (Fig. S5 in the Supplementary Information) and a negative gradient during the nighttime (not shown). For the sparse forests, the temperature gradient is even more complex having a negative or reversed gradient throughout the vertical profiles. By using the current parametrization approach, most of the sparse forest sites required a higher sheer stress (a stronger threshold friction velocity $a_7$) for the within-canopy mixing, compared to dense forest sites (Table S2 in the Supplemen-

tary Information) in order to replicate the measurement results. This observation relates to a general difficulty in being able to simulate canopy transport based on limited general measurements (Stroud et al., 2005).

(3)  Sub-canopy and surface-atmosphere conditions

In this study, we treated the understory and overstory as the same species to construct the vertical $LAI$ profile based on the observed $LAI$ profile. This treatment only allowed the understory growth to follow overstory canopy phenology.

In fact, the forest floor is often occupied by plants with very different traits of which one of the most obvious is the difference in leaf onset and/or leaf fall (Barr et al., 2004). Given the aforementioned model formulation, simulation of the understory phenology and traits could be further improved in the future. For example, overstory and understory vegetation could be simulated as different plant functional types or plant species within the same energy budget column. Also, the microclimate created by the overstory could be used as an input to simulate the environmental conditions in

the understory.

Starting from the point of view of the interaction between ecosystems and the climate, we introduced a weighting factor ($W_{sf}$) as a function of a long-term average temperature, light conditions (gap fraction), transpiration fraction described as $\beta_3$ in the model code and soil evaporation fraction ($\beta_4$) as environmental factors to parametrize surface conductance (Fig. 6) and consequently control the surface latent heat flux. This approach demonstrated the model's capability to simulate





the flux profile in agreement with observations. It may, however, not be valid for the Savanna ecosystem because the understory phenology of this ecosystem relies on water availability in the top soil layer (Baldocchi and Wilson, 2001; Hutley et al., 2000), which is an environmental condition not accounted for in our approach. Furthermore, accounting for ecosystem specific differences in root density profiles and aerial cover of the understory might also help in the simulation

of water and energy fluxes (El Masri et al., 2015; Launiainen et al., 2015). From this perspective, detailed soil moisture profile observations would be very useful in developing a more advanced surface-atmosphere interface parametrization.

(4) The proposed parametrization approach

In general, we provide a simple but useful parametrization approach for the multi-layer energy budget scheme in the global land surface model ORCHIDEE-CAN. Comparing with others studies (Ogée et al., 2003; Staudt et al., 2011;

Launiainen et al., 2015), our approach directly determines the energy and water fluxes and successfully avoids the iterative processes to meet the numerical requirement. In total, a set of twelve parameters need to be prescribed and calibrated regarding the physical processes within the canopy. Our approach presents a good performance at all study sites, though we may have some deficits on wind speed estimation.

## 4.3 Increased model capacity

The innovation of the multi-layer energy and water scheme is the capacity to simulate the behaviour of fluxes within the canopy, and the separation of the soil-level temperature from the temperature of the vegetation levels. The multi-layer scheme helps to address how forest management such as thinning or shelterwood cutting, may alter the forest-atmosphere coupling and resulting fluxes. It also paves the way for the consideration of mixed forests where different plant species or functional types can be in a different microclimatic environment to that of the high-canopy. This capacity is essential for the following types of

applications:

(1) The simulation of emission of biogenic volatile organic compounds (BVOCs), from plants, linking climate change, atmospheric chemistry and the terrestrial biosphere. The implemented multi-layer energy and water budget calculates the leaf temperature and within-canopy radiation, and therefore allows to improve the representation of certain BVOCs, such as isoprene or monoterpene from plants (Guenther et al., 1995, 2006).

(2) Natural disturbances, such as fires, pests and windfall can result in increases in leaf fall, individual tree mortality or complete stand destruction (Lugo, 2008; Seidl et al., 2011; Yue et al., 2014) which in turn determine the vertical $LAI$ profile. The implemented multi-layer energy and water budget scheme calculates the vertical eddy diffusivity and effective drag coefficient as a function of the vertical $LAI$ profile, hence, the new scheme allows the study of effects of changes in disturbance intensity on the energy budget and thus the climate system.

(3) Forest canopy structure plays an important role in regulating the provision of forest ecosystem services such as maintaining biodiversity (Scheffers et al., 2013; Defraeye et al., 2014) or regulating stream flow (Jackson, 2005). Therefore, structural changes to the forest canopy, through, for example, forest thinning or species changes, will reduce the buffer-





ing effect of the canopy. It is only with models including a multi-layer energy budget that an informed prediction of the longterm consequences of land-management policies can be made.

(4) This work takes the first step in exploring the use of vertical canopy profiles in coupled vegetation/atmospheric models, particularly in relation to the calculation of GPP, which is sensitive to the vertical profiles of light, water and nitrogen (Bonan et al., 2012, 2014). To run at a regional or global scale, it is essential to first parametrize the model at the site level.

## 5 Conclusion

Although the first parametrization of a multi-layer energy and water budget scheme did not greatly improve the model performance over the use of the so-called big-leaf approach for energy and water calculations, it provides a more detailed description of the within-canopy micrometeorology of various forest types. A more detailed process description is essential when linking climate change to studies addressing, for example, species vulnerability to climate change, the climate feedbacks from different disturbance intensities, changes in understory habitat following management changes and BVOCs as a result of climate change.

In this study, multiple sites calibration and optimization were performed in order to better understand the functionality of the newly implemented multi-layer energy budget in ORCHIDEE-CAN (revision 2754). Developing the multi-layer energy budget requires accurate field measurements for model calibration and validation. Here we were able to collect and make use of many of the few datasets that exist for intensive in-canopy profile time series measurements. We suggest that more intensive field campaigns, with soil water content observations, especially during the winter season would help in the development of a more reliable parametrization scheme for the within-canopy eddy diffusivity and soil-atmosphere interface conductance. For future model developments, adding an extra soil-atmosphere interface representation such as moss or herbs on the forest floor would be beneficial for a more complete multi-layer energy budget with the objective of describing the surface-atmosphere interface gas and water vapour exchanges.

## 6 Code availability

The code and the run environment are open source. Nevertheless readers interested in running ORCHIDEE-CAN are encouraged to contact the corresponding author for full details and latest bug fixes. The ORCHIDEE-CAN branch with revision 2754 is available via the follow web link (https://forge.ipsl.jussieu.fr/orchidee/browser/branches/ORCHIDEE-DOFOCO/ORCHIDEE?rev=2754)





## 7 Author contributions

YC, JR and SL developed the parametrization scheme. YC, SL and PP designed the study and YC wrote the manuscript with contributions from all co-authors. JR, MJM, JO, KN, SL and AV helped YC with integrating the parametrization scheme for the multi-layer energy budget in ORCHIDEE-CAN. VB and PP provided the optimisation tools and helped with the configuration

5    of these tools. EvG, VH, BH, AK, SLa, DL, EM, JOg, TF and TV provided field observations for all study sites.

*Acknowledgements.* YC, JR, MJM, JO, KN and SL were funded through ERC starting grant 242564 (DOFOCO), and AV was funded through ADEME (BiCaFF).



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





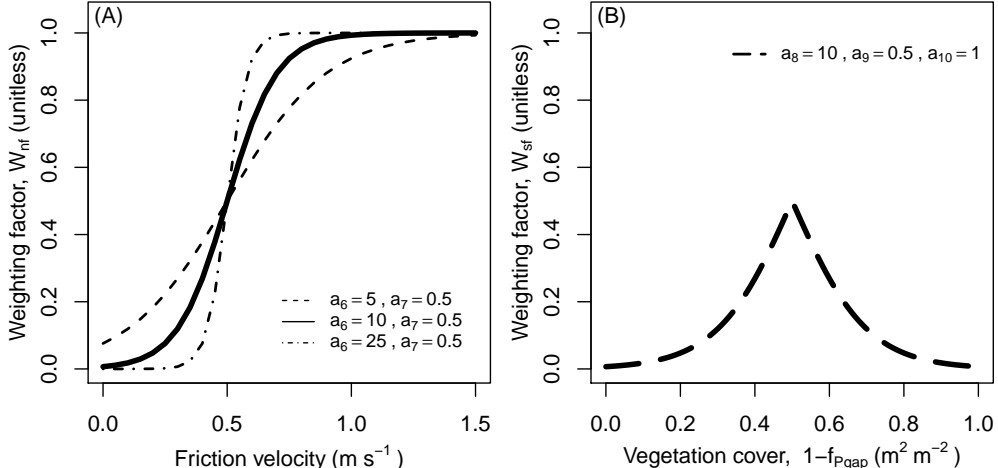

**Figure 1.** Weighting functions for eddy diffusivity and surface conductance. (A) weighting function for the eddy diffusivity ($k$) within the air column (Eq. 3). The weighting is a function of the friction velocity ($u_*$) and was optimized by tuning the parameters $a_6$ and $a_7$. Three different parameter sets show the response of the weighting function to different parameter values. (B) The weighting function for the surface conductance is a function of the vegetation cover (Eq. 7). This weighting function was optimized by tuning the parameters $a_8$ to $a_{10}$. The example has the following parameter values: $a_8$=10.0, $a_9$=0.5, $a_{10}$=1.0 and shows the seasonal cycle of the weights which will be used to scale $k_{surf}$. Values to the left of the deflection point show the effect of an increasing/decreasing overstory cover with an increasing/decreasing temperature in spring/autumn. In spring and autumn understory growth and thus its contribution to evapotranspiration, was assumed to be temperature limited. Values right of the deflection point ($a_9$=0.5) show the dependency of the evapotranspiration on the soil surface layer on the overstory canopy cover when air temperature is no longer limiting understory growth.





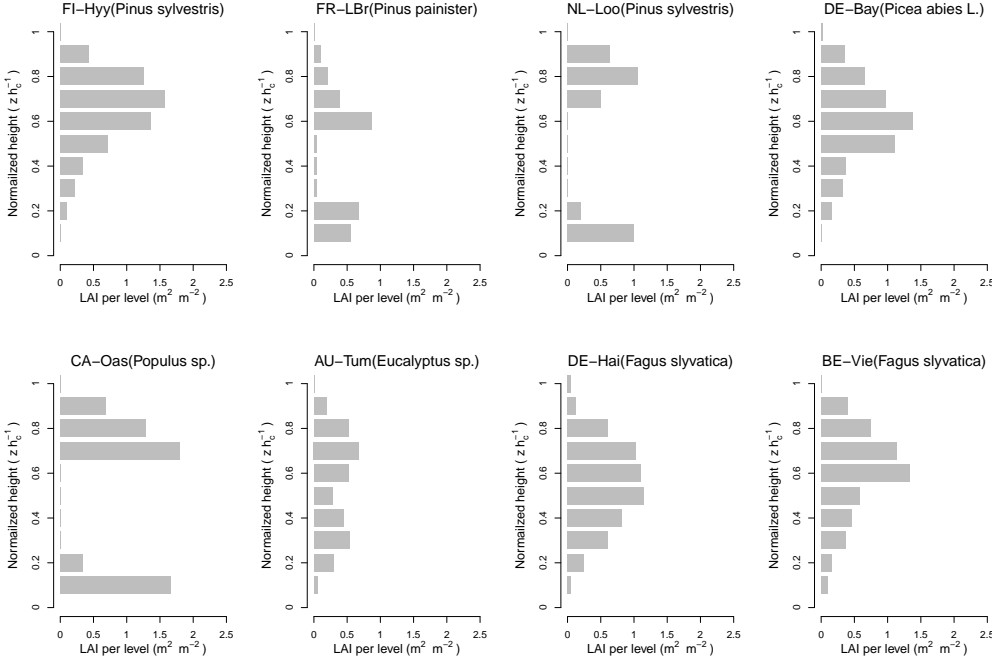

**Figure 2.** Vertical $LAI$ profile for maximal total $LAI$. The $LAI$ was discretized in ten evenly-spaced layers and the canopy height was normalized. The canopies of FI-Hyy, DE-Bay, DE-Hai and BE-Vie were considered dense (Overstory $LAI > 3.0$) whereas the canopies of FR-LBr, NL-Loo, CA-Oas and AU-Tum were considered sparse (Overstory $LAI \leq 3.0$).





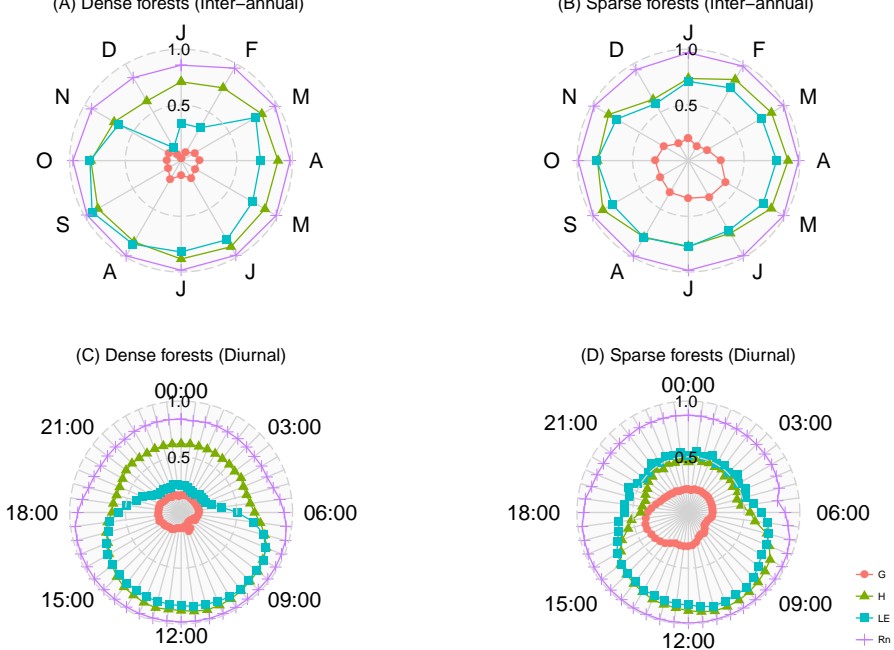

**Figure 3.** Inter-annual and diurnal performance for both dense and sparse forest types, expressed as Taylor skill score ($S_T$), of the single-layer energy budget scheme. Taylor skill score was calculated for each component in the energy budget. Simulations made use of the single-layer energy budget scheme in ORCHIDEE-CAN according to the settings described for experiment 1 (EXP1). Taylor skill scores were aggregated according to canopy density (dense vs. sparse). A value of 1.0 of $S_T$ indicates that model simulations perfectly matches the observations, values lower than 0.5 imply that the model has poor predictive ability.





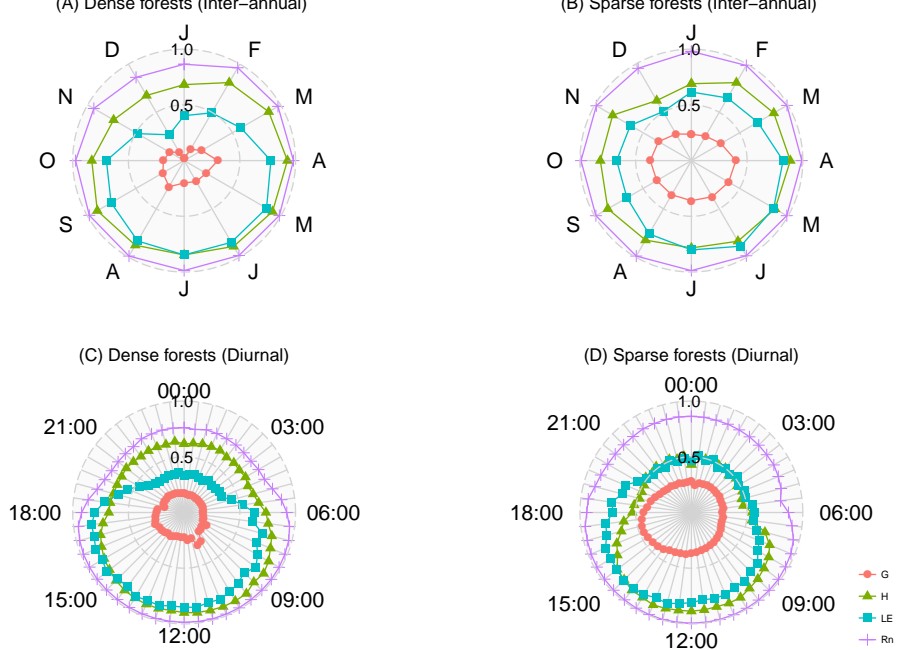

**Figure 4.** Inter-annual and diurnal performance for both dense and sparse forest types, expressed as Taylor skill score ($S_T$), of the multi-layer energy budget scheme. Taylor skill score was calculated for each component in the energy budget. Simulations made use of the multi-layer energy budget scheme in ORCHIDEE-CAN according to the settings described for experiment 3 (EXP3). Taylor skill scores were aggregated according to canopy density (dense vs. sparse). A value of 1.0 of $S_T$ indicates that model simulations perfectly matches the observations, values lower than 0.5 imply that the model has poor predictive ability.





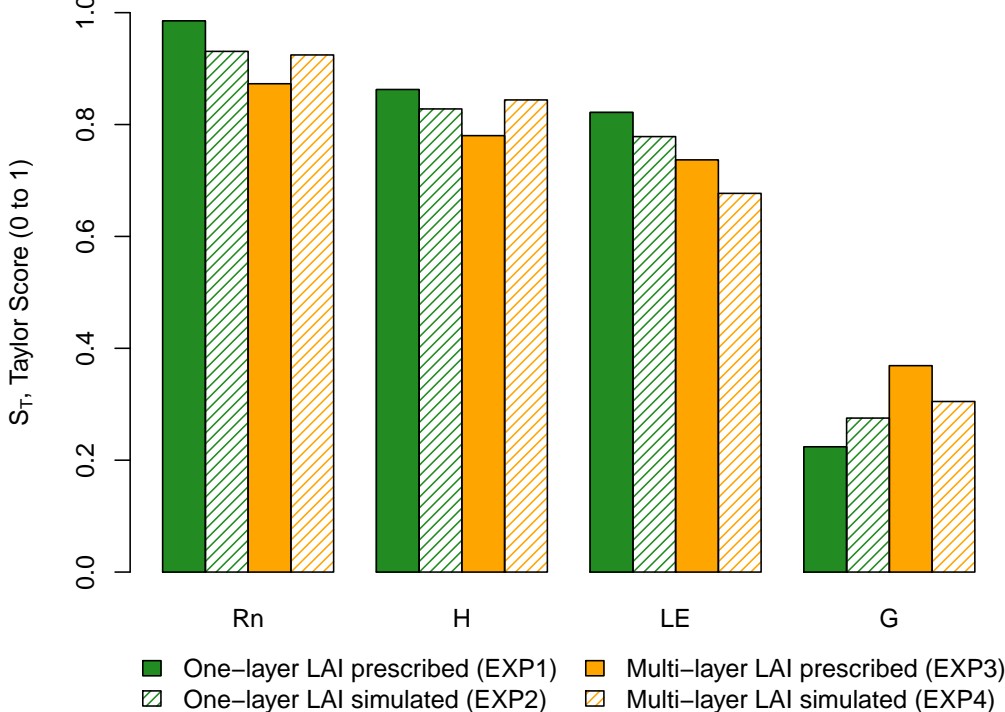

**Figure 5.** Change of model performance, expressed as Taylor skill score, with increasing experimental complexity for both the single-layer and multi-layer energy budget schemes for all eight study sites. EXP1: single-layer scheme with a prescribed $LAI$ profile; EXP2: single-layer scheme with a simulated $LAI$ profile; EXP3: multi-layer scheme with a prescribed $LAI$ profile; EXP4: multi-layer scheme with a simulated $LAI$ profile.





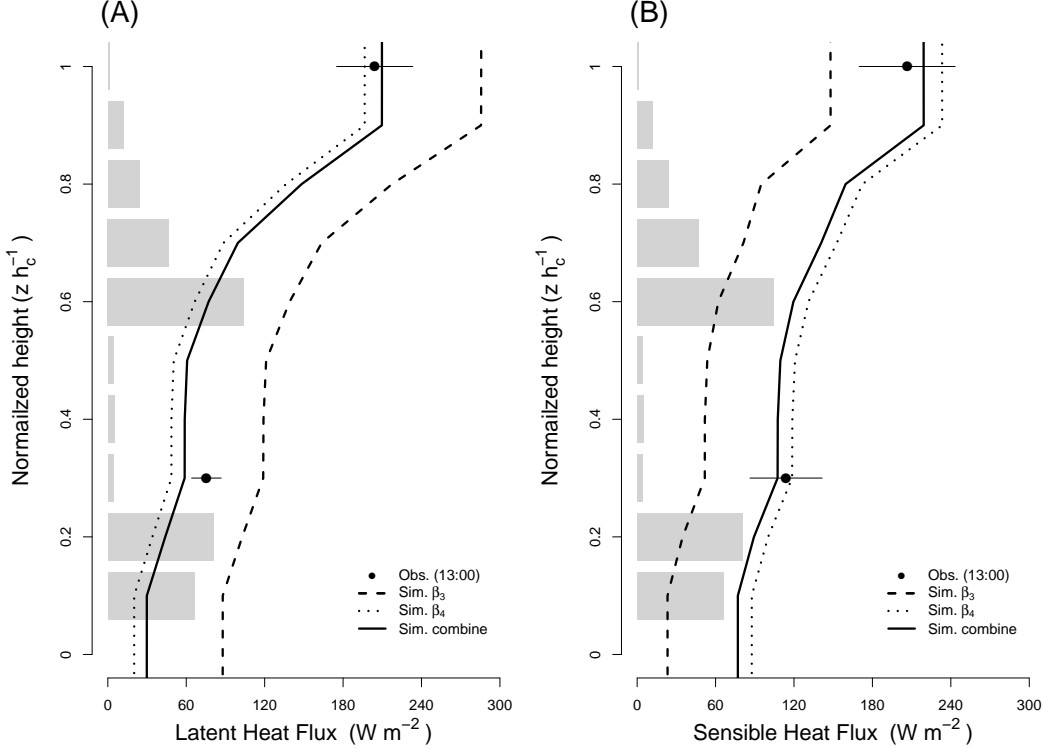

**Figure 6.** Effect of understory phenology on the vertical profile of the latent and sensible heat fluxes at FR-LBr site. (A) Simulated latent heat flux assuming that the interface between the soil and the lowest atmospheric layer behaves as a bare soil (dotted line), a fully vegetated surface (dashed line) or a partly vegetated, partly bare surface where the ratio between bare soil and vegetated soil depends on the understory phenology (full line). The observed profile is shown as black dots where the error bars denote the 5-day temporal variance (B) Simulated sensible heat flux assuming that the interface between the soil and the lowest atmospheric layer behaves as a bare soil (dotted line), a fully vegetated surface (dashed line) or depends on the understory phenology (full line). The observed profile is shown as black dots where the error bars denote the 5-day temporal variance.

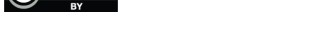

**Table 1.** Symbolic notation used throughout the manuscript

| symbol | description | unit |
|---|---|---|
| $a_1, a_2, a_3, a_4, a_5$ | tuning coefficients for $C_{Deff}$ | unitless |
| $a_6$ | factor ceiling of the slope | unitless |
| $a_7$ | critical friction velocity in the middle point of the S-shape function | unitless |
| $a_8$ | factor to constrain the S-shape function | unitless |
| $a_9$ | threshold for vegetation cover | unitless |
| $a_{10}$ | linear weighting factor | unitless |
| $A$ | assimilation rate | $\mu\,\mathrm{mol\,m^{-2}\,s^{-1}}$ |
| $C_{Deff}$ | effective drag coefficient | unitless |
| $C_S$ | concentration of $CO_2$ at leaf surface | ppm |
| $C_{D,i}$ | vertically discretised estimate for canopy drag coefficient | unitless |
| $D_{h,air}$ | heat diffusivity of air | $\mathrm{cm^2\,s^{-1}}$ |
| $D_{h,H_2O}$ | heat diffusivity of water vapour | $\mathrm{cm^2\,s^{-1}}$ |
| $d_l$ | characteristic leaf length | m |
| $f_{Pgap}$ | over-story gap probability from P gap fraction | $\mathrm{m^2\,m^{-2}}$ |
| $g_0$ | residual stomatal conductance if the irradiance approaches zero | $\mathrm{m\,s^{-1}}$ |
| $h_s$ | relative humidity at leaf surface | % |
| $h_c$ | canopy height | m |
| $k_i$ | diffusivity for level $i$ | $\mathrm{m^2\,s^{-1}}$ |
| $k_i^*$ | modified diffusivity for level $i$ | $\mathrm{m^2\,s^{-1}}$ |
| $k_{surf}$ | conductance for the surface-atmosphere interface | $\mathrm{m\,s^{-1}}$ |
| $LAI_i$ | leaf area index at level $i$ | $\mathrm{m^2\,m^{-2}}$ |
| $Nu$ | Nusselt number | unitless |
| $P_{m,i}$ | momentum shielding factor | unitless |
| $PAI$ | plant area index | $\mathrm{m^2\,m^{-2}}$ |
| $R$ | correlation coefficient between the simulation and the observation | unitless |
| $R_0$ | maximum correlation coefficient | unitless |
| $R_{b,i}$ | boundary layer resistance at level $i$ for heat | $\mathrm{s\,m^{-1}}$ |
| $R'_{b,i}$ | boundary layer resistance at level $i$ for water vapour | $\mathrm{s\,m^{-1}}$ |
| $R_{s,i}$ | stomatal resistance at level $i$ | $\mathrm{s\,m^{-1}}$ |
| $Re$ | Reynold's number | unitless |
| $SLA$ | specific leaf area | $\mathrm{m^2\,g^{-1}}$ |
| $S_T$ | Taylor skill score | unitless |
| $\overline{T}_a$ | mean air temperature during the last 21 days | K |
| $T_L$ | Lagrangian timescale | s |
| $u_*$ | friction velocity | $\mathrm{m\,s^{-1}}$ |
| $u_i$ | velocity at level $i$ | $\mathrm{m\,s^{-1}}$ |
| $V_{cmax}$ | carboxylation capacity | $\mu\,\mathrm{mol\,m^{-2}\,s^{-1}}$ |





**Table 1.** Continuation of Table 1

| symbol | description | unit |
| --- | --- | --- |
| $W_{br}$ | weighting parameter for boundary layer resistance | unitless |
| $W_{nf}$ | near-field weighting factor | unitless |
| $W_{sf}$ | weighting parameter for atmosphere-surface conductance | unitless |
| $W_{sr}$ | linear reduction parameter for stomatal resistance | unitless |
| $\beta_3$ | fraction of potential plant transpiration realized | unitless |
| $\beta_4$ | fraction of soil evaporation realized | unitless |
| $\mu$ | kinematic viscosity of air | $\mathrm{cm^2\,s^{-1}}$ |
| $\hat{\sigma}_f$ | ratio of the variance of the simulations over the variances of observations | unitless |
| $\sigma_w$ | standard deviation in vertical velocity | $\mathrm{m\,s^{-1}}$ |

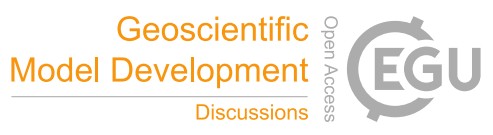
**Table 2.** Stand structure and data availability of the experimental sites. The maximum observed leaf area ($LAI$; m² m⁻²) of the overstory and understory $LAI$ (all-sided) are reported separately. Height of the overstory is reported in m. $U$ denotes wind speed, $T_a$ denotes atmospheric temperature and $q_a$ denotes atmospheric humidity. $LE$, $H$ and $R_n$ denote the latent heat flux, the sensible heat flux and the net radiation, respectively. + indicates that profile measurements were available. - indicates that no profile measurements were available.

| Site Code | FI-Hyy | FR-LBr | NL-Loo | DE-Bay | CA-Oas | AU-Tum | DE-Hai | BE-Vie |
|---|---|---|---|---|---|---|---|---|
| Species | Pinus sylvestris | Pinus pinaster | Pinus sylvestris | Picea abies | Populus sp. | Eucalyptus sp. | Fagus sylvatica | Fagus sylvatica* |
| Leaf type | Needleleaved | Needleleaved | Needleleaved | Needleleaved | Broadleaved | Broadleaved | Broadleaved | Broadleaved |
| Growth form | Evergreen | Evergreen | Evergreen | Evergreen | Deciduous | Evergreen | Deciduous | Mixed |
| ORCHIDEE PFT | 18 | 5 | 6 | 7 | 20 | 15 | 13 | 13 |
| Overstory $LAI$ | 6.5 | 2.0 | 1.9 | 4.8 | 2.9 | 2.5 | 5.8 | 5.1 |
| Understory $LAI$ | 0.5 | 1.5 | 1.5 | 0.5 | 2.8 | 1.0 | 0.1 | 0.1 |
| Height | 17.0 | 23.0 | 15.0 | 15.0 | 22.0 | 50.0 | 30.0 | 25.0 |
| $U$ profile | + | – | + | + | + | + | + | + |
| $T_a$ profile | + | + | + | + | + | + | + | + |
| $q_a$ profile | + | + | + | + | + | + | – | + |
| $LE$ profile | + | + | + | +** | + | – | – | – |
| $H$ profile | + | + | + | + | + | + | + | – |
| $R_n$ profile | – | + | + | +** | – | – | – | – |
| Reference | (Launiainen et al., 2007) | (Ogée et al., 2003; Porte et al., 2000) | (Dolman et al., 2002; Moors, 2012) | (Foken et al., 2012; Staudt et al., 2011) | Barr et al. (2004) | (Haverd et al., 2012; Lovell et al., 2012) | (Knohl et al., 2003) | (Aubinet et al., 2001; Laitat et al., 1998) |

*: This site is partially mixed with *Pseudotsuga menziesii*

**: $LE$ profile was available for 2007 and 2008 period but not 2011, and $R_n$ profile was partly available in 2007



**Table 3.** Observation periods for the different data uses in this study. Date format: dd/mm/yy. The information of the energy closure gap for each site over different selected periods was also calculated based on Chen and Li (2012)). EXP1: single-layer scheme with a prescribed $LAI$ profile; EXP2: single-layer scheme with long-term a simulated $LAI$ profile; EXP3: multi-layer scheme with a prescribed $LAI$ profile; EXP4: multi-layer scheme with a simulated $LAI$ profile.

| Site Code | FI-Hyy | FR-LBr | NL-Loo | DE-Bay | CA-Oas | AU-Tum | DE-Hai | BE-Vie |
|---|---|---|---|---|---|---|---|---|
| Period for short-term parameters optimization (**Period I**) | 01/08/06 14/08/06 | 31/07/06 05/08/06 | 08/07/97 12/07/97 | 04/07/11 17/07/11 | 16/08/94 22/08/94 | 08/11/06 11/11/06 | 10/05/01 19/05/01 | 01/08/02 07/08/02 |
| Closure gap ($Wm^{-2}$) | 43.34 | 41.56 | 10.48 | 18.97 | 19.82 | 18.40 | 29.89 | 28.19 |
| Period for long-term parameters optimization (**Period II**) | 01/01/02 31/12/02 | 01/01/03 31/12/03 | 01/01/02 31/12/02 | 01/01/97 31/12/97 | 01/01/05 31/12/05 | 01/06/01 31/06/02 | 01/01/05 31/12/05 | 01/01/97 31/12/97 |
| Closure gap ($Wm^{-2}$) | 11.47 | 21.59 | 15.38 | 42.47 | 2.89 | 7.12 | 27.83 | 42.43 |
| Period for single-year EXP1 and EXP3 validation (**Period III**) | 01/01/05 31/12/05 | 01/01/06 31/12/06 | 01/01/97 31/12/97 | 01/01/99 31/12/99 | 01/01/04 31/12/04 | 01/06/04 31/06/05 | 01/01/01 31/12/01 | 01/01/02 31/12/02 |
| Closure gap ($Wm^{-2}$) | 10.99 | 13.20 | 16.61 | 50.24 | 4.13 | 7.73 | 23.49 | 42.43 |
| Period for multi-year EXP2 and EXP4 validation (**Period IV**) | 01/01/02 31/12/06 | 01/01/03 31/12/06 | 01/01/02 31/12/06 | 01/01/97 31/12/99 | 01/01/04 31/12/05 | 01/06/01 31/06/05 | 01/01/00 31/12/06 | 01/01/97 31/12/06 |
| Closure gap ($Wm^{-2}$) | 10.68 | 17.03 | 22.65 | 48.14* | 3.51 | 9.40 | 23.69 | 33.77 |

*: The forest was 1997-99 strongly affected by forest decline, 2011 the forest was again in a good state



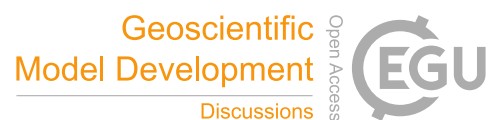

**Table 4.** Description of parameters, code reference, initial values and tuning ranges used in the multi-layer energy budget model in this work.

| Symbol | Description | ORCHIDAS name | Default value | Tuning range |
|---|---|---|---|---|
| $a_1$ | parameter for tuning layer dynamic drag coefficient ($C_{Deff}$) | a_1 | 6.410 | use default |
| $a_2$ | parameter for tuning $C_{Deff}$ | a_2 | 0.001 | use default |
| $a_3$ | parameter for tuning $C_{Deff}$ | a_3 | 0.434 | 0.1 to 0.8 |
| $a_4$ | parameter for tuning $C_{Deff}$ | a_4 | -0.751 | -0.9 to -0.1 |
| $a_5$ | parameter for tuning $C_{Deff}$ | a_5 | 0.071 | 0.05 to 0.1 |
| $a_6$ | parameter for tuning eddy diffusivity ($W_{nf}$) | k_eddy_slope | 5.0 | 1.0 to 20.0 |
| $a_7$ | parameter for tuning $W_{nf}$ | k_eddy_ustar | 0.3 | 0.0 to 0.6 |
| $a_8$ | parameter for tuning surface-atmosphere interface conductance ($W_{sf}$) | ks_slope | 5.0 | 1.0 to 20.0 |
| $a_9$ | parameter for tuning $W_{sf}$ | ks_veget | 0.5 | 0.0 to 1.0 |
| $a_{10}$ | parameter for tuning $W_{sf}$ | ks_tune | 1.0 | 0.5 to 1.5 |
| $W_{br}$ | weighting factor for tuning layer boundary resistance | br_fac | 1.0 | 0.1 to 10.0 |
| $W_{sr}$ | weighting factor for tuning layer stomatal resistance | sr_fac | 1.0 | 1.0 to 10.0 |