# Peer review of "Evaluating the performance of the land surface model ORCHIDEE-CAN on water and energy flux estimation with a single- and a multi- layer energy budget scheme"

_Geoscientific Model Development, 2016_

## Referee Comment (RC1) · Anonymous Referee #1 · 10 Mar 2016

This paper deals with the improvement of an existing SVAT by application of a multi-layer vegetation approach. In general it is a good idea to test models with higher complexity for application in "reality". So the manuscript has a good approach for publication. But, apart from the discussion about the performance of the new approach in relation to the former approach, some points must be taken into account for the acceptance of the paper:

Equation 7 uses the threshold of 298.15 K. What is the physical basis for this threshold - or is is an empirical value?

Equation 11 describes the calculation of stomata resistance dependent on photosynthesis activity of the plant (Farquhar model). This leaf photosynthesis model does not consider interaction between stomata resistance and soil water availability (stomata regulation by trees in case of disturbed water supply from soil). The authors must check the literature to include this effect in eq. 11 (e.g., literature about stomata control and decoupling coefficient).

The authors should explain how they want to tackle the mismatch between rough resolution of driving data (reanalysis 0.5 degree) and high vertically resolved vegetation layer. Is it necessary in this case to leave the bigleaf concept? Apart from that, it is doubtful whether reanalysis data with a resolution of 0.5x0.5 degree give a realistic information for soil water pool.

The performance of the model strongly depends on model tuning. There are a couple of tuning parameters without plausible natural background. This fact makes a transferability of the results to other sites difficult. Could the authors discuss this problem?

The multi-layer approach shows an improvement especially in soil heat flux. Is it relevant for climate? Apart from that, for inter-annual cycle soil heat flux must be about zero (not fulfilled in Fig. 4)!

I recommend a major revision of the paper and I would review the paper again.

---

## Referee Comment (RC2) · Anonymous Referee #2 · 6 Apr 2016

Chen et al.: Evaluating the performance of the land surface model ORCHIDEE-CAN on water and energy flux estimation with a single- and a multi- layer energy budget scheme

Ryder et al. (2016) described a new canopy parameterization for the ORCHIDEE land surface model that allows for a multi-layer canopy. Here, Chen et al. apply that model at 8 forests sites of different species composition, height, and leaf area index. They compare the model to observations at the site, both of fluxes above the canopy and profiles of temperature and wind speed within the canopy, and also compare the model

to a standard big-leaf canopy model. These comparisons show that multi-layer canopy models are a viable path forward and can be used in land surface models. The paper makes recommendations about further research needs. These are all important points and will help advance the land surface modeling community.

1. My primary concern with the manuscript is that the model has 10 or 12 free parameters that the authors optimized by fitting the model results to the observations at each site. These parameters lack a physical basis and are in effect tuning knobs. The optimization procedure produced significant improvement compared with the non-optimized parameters. This fitting of the model to the data does not test the theory in the model. The model uses the second-order closure model of Massman and Weil (1999) to calculate the vertical diffusivity. The Massman and Weil model has not been widely used. How robust is the theory? The authors introduce a weighting factor that modifies the diffusivity based on friction velocity (not in the Massman and Weil model). What is the basis for this? The authors also calculate the canopy drag coefficient using a parameterization developed by Wohlfahrt and Cernusca (2002) for grassland. Should we expect this to work in forests? It is important to note that Massman and Weil used a different parameterization for the drag coefficient and did not have the weighting factor. The use of numerous free parameters to fit the model to the observations obscures whether these parameterizations are theoretically sound and applicable to forests. The authors acknowledge this with the statement that "a set of twelve parameters need to be prescribed and calibrated regarding the physical processes within the canopy" (page 16, line 11). One is left wondering how robust the parameterization of physical processes is given this many parameters used to tune the model.

2. The vertical diffusivity ($k_i$) is described by equations (3) and (6), which are different. Which one is used to calculate $k_i$? How does equation (6) relate to equation (3). How is the Lagrangian timescale ($T_{Li}$) in equation (3) calculated? More generally, where does equation (6) come from? I do not see it in either the Ryder et al. (2016) paper that describes the model or the Haverd et al. (2012) paper that is given as a reference.

3. Line 13, page 6: Deff should be CDeff

4. Page 7: Explain how ksurf is used in the model.

5. Figures 3 and 4 are nice summaries of overall model performance, but it is unclear how the Taylor scores relate to the magnitude of biases. Sensible heat flux and latent heat flux have low Taylor scores at particular times of the year or times of the day. It would be helpful to have plots of model and observed fluxes for both the annual cycle and the diurnal cycle so that the reader can clearly see the magnitude of the flux biases.

---

## Referee Comment (RC3) · Anonymous Referee #1 · 21 Apr 2016

Dear Authors, thank your for the constructive answers to my comments. I am waiting for a revised manuscript version to make final comments.

---

## Author Response (AR1)

Dear Editor,

We would like to thank you for the effort in reviewing this study and the opportunity to prepare a revised manuscript. In line with our reply to the discussion, we implemented all except one, of the suggestions and concerns of the reviewers. The exception was comment 1 made by referee 1. Although we were able to address the reviewer's concern by changing the equation, the changes were different from those anticipated in the reply. However, the revised approach tries to better describe the dynamics of the under-story phenology, as suggested by the reviewer. As a consequence of these changes we had to re-run the model which resulted in revisions for figures 1, 4, 5, 6 and 7 and tables 1, 4, S2, S3 and S4.

Below we tabulate the link between the discussion and the revised manuscript by summarizing the changes made to the manuscript.

| Referee No. | Comment No. | Page | Lines | Figure No. | Table No. |
|---|---|---|---|---|---|
| #1 | #1 | 7 | 26 to 30 | 1 and 4 | 1, S2, S3 and S4 |
| | | 8 | 1 to 5, 9 to 17, 26 to 28 | | |
| | #2 | 9 | 17 to 28 | | |
| | #3 | 12 | 14 to 24 | | |
| | #4 | 18 | 4 to 15 | | |
| | #5 | 16 | 8 to 15 | | 4 |
| | | 18 | 21,22, 24 to 28 | | |
| | #6 | 15 | 10 to 12 | 5 and 7 | |
| #2 | #1-1 | 3 | 13, 15 to 22 | | |
| | | 16 | 8 to 15 | | |
| | | 18 | 21,22, 24 to 28 | | 4 |
| | #1-2 | 6 | 21 to 25 | | |
| | | 7 | 1 to 4, 23 to 25 | | |
| | #1-3 | 5 | 27 to 32 | | |
| | | 6 | 1 to 6 | | |
| | #2 | 7 | 23 to 25 | | |
| | #3 | 7 | 5 | | |
| | #4 | 7 | 27 to 30 | | |
| | | 8 | 1 and 4 | | |
| | #5 | 14 | 27, 29, 30 and 33 | 5 and 6 | |
| | | 15 | 1 to 12, and 14 | | |

I look forward to hearing from you at your earliest convenience.

Kind Regards,

Yi-Ying Chen on behalf of the author team

Postdoctoral Research Fellow

Laboratoire des Sciences du Climat et de l'Environnement, LSCE/IPSL,

CEA-CNRS-UVSQ, Université Paris-Saclay, F-91191 Gif-sur-Yvette, France

Now at

Graduate Institute of Hydrological and Oceanic Sciences

National Central University, Taiwan

Tel: +886-928-299469

spancer_hot@hotmail.com

Referee#1

We would like to thank both reviewers for their insightful comments. Below we discuss how we will address their concerns in the revised manuscript.

**#1 Equation 7 uses the threshold of 298.15 K. What is the physical basis for this threshold - or is is an empirical value?**

We would like to thank the reviewer for pointing out the aforementioned issue, i.e. "The threshold of 298.15 K may be only suitable for sites in the temperate climate zone (with temperate grass species)". Indeed, this threshold temperature should reflect the geographical variation for different sites or locations. To the extent of the current approach to global applications, the generic temperature of 298.15 K will need to be replaced by a localized threshold.

Equation 7 describes the seasonality of the soil-atmosphere interface, which we believe is driven by the under-story and its phenology (Launiainen et al., 2015). Currently, the model does not simulate the production nor the phenology of the under-story. As a substitute for this rather complex process, we made use of a weighting coefficient for the conductance of the soil-atmosphere interface ($K_{surf}$) or, in other words, the calculation of the water vapor exchange between the soil layer and the first air column ($\Phi_{\lambda E}$) (see the $\Phi_{\lambda E}$ and $K_{surf}$ in the figure below and the formal description of using $K_{surf}$, which is given in the supplementary material of Ryder et al. (2016), in Equation S4.30 and S4.31).

[Figure]

In Equation 7, we used 298.15 K as a threshold to simulate over-story phenology. Above this threshold, we use the sum of the canopy gaps as a proxy for the under-story phenology. In other

words, the current approach assumes that when the long-term (21 days) mean t2m temperature exceeds 15°C (298.15 K), shading from the over-storey will become the main driver over the under-story phenology. Given the spatial distribution of our study sites, this is a crude but defendable assumption.

As an intermediate solution between this validation exercise and the global application in the next study, we will search for a more general parameterization of this threshold temperature and we will try to modify the reference temperature in Equation 7 by using a global soil temperature map instead. This, implies that we will have to rerun the model optimization work for the tuning coefficients $a_8$ to $a_{10}$.

**#2 Equation 11 describes the calculation of stomata resistance dependent on photosynthesis activity of the plant (Farquhar model). This leaf photosynthesis model does not consider interaction between stomata resistance and soil water availability (stomata regulation by trees in case of disturbed water supply from soil).**

The reviewer expressed concern for the absence of soil water availability in the calculation of stomatal resistance in Equation 11. After re-reading the text we understand where this concern originates, but our model formulation accounts for soil water stress in the calculation of actual transpiration and in turn in stomatal conductance and photosynthesis. ORCHIDEE-CAN calculates the supply of the water available for transpiration ($F_{Trs}$) as the pressure difference between the soil and the leaves ($p_{delta}$) divided by the sum of hydraulic resistances of fine roots ($R_r$), sapwood ($R_{sap}$) and leaves ($R_l$), i.e., $F_{Trs}=p_{delta}/(R_r+R_{sap}+R_l)$ (see Equation 20 in Naudts et al., 2015). The atmospheric demand of water for transpiration is calculated as the vapor pressure difference between the leaves and atmosphere divided by the sum of boundary layer resistance ($R_b$) and stomatal resistance ($R_s$) (see Equations 9, 14 and 15 in Ryder et al., 2016). When the supply can satisfy the demand, there is no water stress and photosynthesis ($A$) is calculated. When the demand is limited by the supply term, $A$ and $R_s$ are recalculated such that they satisfy the supply. Water stress thus enters Equation 11 in the value of $A$. Through Equation 11, we add a weighting factor ($W_{sr}$) to the original calculation of stomatal resistance ($R_s$) to tune the final calculation of the transpiration demand term (this tuning factor represents the coupling of the canopy to the atmosphere). Following the above reasoning, we will improve the description of equation 11 to eliminate the misunderstanding concerning how ORCHIDEE-CAN accounts for soil water stress.

**#3 The authors should explain how they want to tackle the mismatch between rough resolution of driving data (reanalysis 0.5 degree) and high vertically resolved vegetation layer. Is it necessary in this case to leave the bigleaf concept?**

Using forcing data of a rough spatial resolution to drive the model may contain information derived from several different land cover types, thus this comment touches upon an interesting issue: how to account for the average surface fluxes from the contribution of different subgrid scale land cover types? The present ORCHIDEE single-layer model calculates a weighted average of different PFTs across a grid square to calculate a total representative flux. An alternative approach, and one that we are investigating using this multi-layer model, is to calculate the heat fluxes of each vegetation type separately (sub-grid scale modeling) so that the mixing occurs above the canopy. We will add this point to the discussion.

**#4 Apart from that, it is doubtful whether reanalysis data with a resolution of 0.5x0.5 degree give a realistic information for soil water pool.**

For the spin-up of the initial state of the soil water pool, 20 years of climate data are required. We had a choice between using local high resolution climate observations for a usually very limited time period or using low resolution regional re-analysis for a much longer time period. Using the local high resolution data would have the advantage that local information is used, but due to the fact that some time series are only 2 to 4 years long (**Table 3** Period IV in Chen et al.), the spin-up would have to cycle 5 to 10 times over the same data. Although local data could then still have been used, cycling gives a lot of weight to the climatic events in the time series and may as such result in a biased spin-up. The alternative is to use 20 years of a climate re-analysis, these data represent the inter-annual variability better than cycling over the same 2 or 4 years of data but has the disadvantage that the data are less likely to represent the local conditions (especially in mountainous regions). Given the fact that we did not have access to soil water content data, we could not evaluate which method is better to spin-up the soil water content in the model. For this reason, we performed a sensitivity analysis of the parameterization of the initial soil water content at one of the driest sites used in this study (In the section 3.1 Model parameterization: Page 12 Line 23-25 and **Fig. S7** in the supplementary information from Chen et al.). Note that the model calibration and validation were based on the site level observations because that part of the study did not require cycling of the same data. In short, in the absence of a rigorous validation of both approaches to the spin-up of the soil water content, it is not possible to rank one method above the other. In the revised text we will clarify the strengths and weaknesses of the two present different approaches.

**#5 The model performance strongly depend on the model tuning. There are a couple of tuning parameters without plausible natural background. This fact makes a transferability of the results to other sites difficult. Could the authors discuss this problem?**

This comment refers to a long-standing issue in model development and model validation which is very well discussed by Oreskes et al. (1994). Despite the direction of the land surface model community towards the development of more mechanistic models, all large-scale land surface models contain an important level of empiricism. When the model is carefully developed and validated the empirical parameters mimic an overly complex (for the purpose of the model) or poorly understood process. As we tried to follow this philosophy we believe that our parameters have a plausible natural background but this does not overcome the issue of equifinality of the model. Ideally, future developments should aim at replacing such parameters by a more mechanistic approach if the empirical module represents a process that is at the core of the objectives of the model.

| Tuning parameter names used in this study | Physical parameter | Empirical representation of |
|---|---|---|
| $a_1$ to $a_5$ | effective surface drag | Bending of tree branches to increase the contact surface |
| $a_6$ to $a_7$ | eddy diffusivity | Inner canopy turbulent mixing induced by canopy structure |
| $a_8$ to $a_{10}$ | surface-atmosphere conductance | Sub-canopy phenology |

| $W_{br}$ | layer boundary resistance | Upscaling the atmospheric coupling for the heat transfer from a single leaf to the entire canopy |
|---|---|---|
| $W_{sr}$ | layer stomatal resistance | Upscaling the atmospheric coupling for the water vapor transfer from a single leaf to the entire canopy |

In Ryder et al. 2016, the model was developed and tested for a single site. In the current manuscript we aim to test the model for more diverse environmental conditions in order to demonstrate that the numerics can deal with the variation that can be found in global ecosystems. For this we granted ourselves the freedom to derive a separate parameter set for each site. By doing so we learned about the strengths and weaknesses of the model and its parameters. Next, we will have to derive a single parameter set for each PFT and test how well the model reproduces global patterns in, for example, evapotranspiration. This is the point of the development and validation chain, where we will learn about the transferability of the parameters. We will address this issue in the manuscript by rephrasing parts of the introduction and adding a paragraph to the discussion.

**#6 The multi-layer approach shows an improvement especially in soil heat flux. Is it relevant for climate? Apart from that, for inter-annual cycle soil heat flux must be about zero (not fulfilled in Fig. 4)!**

Comparing the observed magnitude of soil heat flux with other components of the surface energy budget shows that at forest sites the soil heat flux is almost one order of magnitude smaller than the other components. The reported result - that the multi-layer simulation shows a better model prediction skill is interesting (as discussed), but is unlikely to be sufficient to justify the added complexity of a multi-layer model. However, the soil heat flux is an essential aspect in simulating the snow phenology (Wang et al., 2015). Therefore, improved simulations of the soil heat fluxes could have important indirect effects on climate simulations of regions with a pronounced snow season.

The reviewer remarks that the inter-annual cycle of soil heat flux should be zero. This is indeed to be expected for graphs showing the absolute soil heat flux. **Fig. 4**, however, shows the model skill for different components in the energy budget – the annual sum of the model skill should not be zero. We will prepare new figures showing the absolute values for both the observations and simulations at the diurnal and inter-annual scale.

Referee#2

We would like to thank both reviewers for their insightful comments. Below we discuss how we will address their concerns in the revised manuscript.

**#1 My primary concern with the manuscript is that the model has 10 or 12 free parameters that the authors optimized by fitting the model results to the observations at each site. These parameters lack a physical basis and are in effect tuning knobs. The optimization procedure produced significant improvement compared with the nonoptimized parameters. This fitting of the model to the data does not test the theory in the model. The model uses the second-order closure model of Massman and Weil (1999) to calculate the vertical diffusivity. The Massman and Weil model has not been widely used. How robust is the theory? The authors introduce a weighting factor that modifies the diffusivity based on friction velocity (not in the Massman and Weil model). What is the basis for this? The authors also calculate the canopy drag coefficient using a parameterization developed by Wohlfahrt and Cernusca (2002) for grassland. Should we expect this to work in forests? It is important to note that Massman and Weil used a different parameterization for the drag coefficient and did not have the weighting factor. The use of numerous free parameters to fit the model to the observations obscures whether these parameterizations are theoretically sound and applicable to forests. The authors acknowledge this with the statement that "a set of twelve parameters need to be prescribed and calibrated regarding the physical processes within the canopy" (page 16, line 11). One is left wondering how robust the parameterization of physical processes is given this many parameters used to tune the model.**

**- The authors optimized by fitting the model results to the observations at each site. These parameters lack a physical basis and are in effect tuning knobs. The optimization procedure produced significant improvement compared with the nonoptimized parameters. This fitting of the model to the data does not test the theory in the model.**

With regards to this comment, a similar observation is made by referee #1 (comment #5) and refers to a long-standing issue in model development and model validation which is very well discussed by Oreskes et al. (1994). Despite the ambitions of the land surface model community to move towards more mechanistic models, all large-scale land surface models contain an important level of empiricism. When the model is carefully developed and validated the empirical parameters mimic an overly complex (for the purpose of the model) or poorly understood process. As we tried to follow this philosophy, we believe that our parameters have a plausible basis but this does not overcome the issue of equifinality of the model. Ideally, future developments should aim at replacing such parameters by a more mechanistic approach if the empirical module represents a process that is at the core of the objectives of the model.

| Tuning parameter names used in this study | Physical parameter | Empirical representation of |
|---|---|---|
| $a_1$ to $a_5$ | effective surface drag | Bending of tree branches to increase the contact surface |
| $a_6$ to $a_7$ | eddy diffusivity | Inner canopy turbulent mixing induced by canopy structure |
| $a_8$ to $a_{10}$ | surface-atmosphere conductance | Sub-canopy phenology |

| $W_{br}$ | layer boundary resistance | Upscaling the atmospheric coupling for heat transfer from a single leaf to the entire canopy |
|---|---|---|
| $W_{sr}$ | layer stomatal resistance | Upscaling the atmospheric coupling for vapor transfer from a single leaf to the entire canopy |

**- The model uses the second-order closure model of Massman and Weil (1999) to calculate the vertical diffusivity. The Massman and Weil model has not been widely used. How robust is the theory? The authors introduce a weighting factor that modifies the diffusivity based on friction velocity (not in the Massman and Weil model). What is the basis for this?**

This is the first attempt for the implementation of the multi-layer energy budget in ORCHIDEE-CAN, and we seek an analytical physical model to calculate the wind profile from the canopy top down to the ground level. In the initial phase (Ryder et al., 2014), we attempted a validation of the original model by using in-situ observation scalar profiles at a single site. We found that there was a bias in the estimation of the air temperature profile within the canopy layer during nighttime (see Page 8674, line 4 to line 19 in Ryder et al., 2014. These issues have been well-documented in the scientific literatures (Gao et al., 1989; Dolman and Wallace, 1991; Makar et al., 1999; Wolfe and Thornton, 2010). One possible, although empirical, solution is to adjust the simulated eddy diffusivity by using a factor dependent on the state of turbulent mixing, which was proposed in this study (see Equation 5 in this manuscript). After completion of the current site level validation work, we were able to better understand the capability and sensitivity of the parameters used in the model. Future studies may focus on replacing this empirical solution by a more mechanistic solution. In the context of ORECHIDEE and its coupling to the atmospheric model, this implies that we will have to search for an implicit solution of the near-field far-field theory by Raupach (1989).

**- The authors also calculate the canopy drag coefficient using a parameterization developed by Wohlfahrt and Cernusca (2002) for grassland. Should we expect this to work in forests?**
The canopy structure is a very important characteristic for the land-atmosphere interaction, which can now be simulated by the land surface model OCHIDEE-CAN. We assumed that the drag coefficient is scalar independent and can be parametrized by the canopy structure. The effective drag coefficient used in the MW1999 model is assumed to be a constant throughout the canopy layer, but it also can be treated as a function of the vertical canopy structure. In this study, we made use of a prototype parameterization approach proposed by Wohlfahrt and Cernusca (2002). Wohlfahrt and Cernusca provided the basic idea for considering the effective drag coefficient, that can be varied due to changes of canopy structure, such as bending effects. Thus, we adopted this parametrization to our model; however we left the first two tuning coefficients ($a_1$ and $a_2$) as constant. This modification allows the effective drag to reduce from a large value to a constant while moving from the top of the canopy to the soil surface layer. Thus, we didn't apply exactly the surface drag parameterization for grasses. More precisely, we applied the ideas derived in grassland research to a forest canopy. We will address this issue in the revised manuscript.

**#2 The vertical diffusivity (ki) is described by equations (3) and (6), which are different. Which one is used to calculate ki? How does equation (6) relate to equation (3). How is the Lagrangian timescale (TLi) in equation (3) calculated? More generally, where does equation (6) come from? I do not see it in either the Ryder et al. (2016) paper that describes the model or the Haverd et al. (2012) paper that is given as a reference.**

We would like to thank the reviewer for drawing our attention to this problem. Firstly we cited the wrong paper: the correct reference is Haverd et al. in 2009, published in the boundary layer meteorology. Secondly, we did not well explain the transition from equation 3 to 6.

There exists a variety of parameterization approaches, of which the most simple is to assume a constant value between 0.25 to 0.4 or a linear function that decreases to zero when moving into the canopy layer. Here, we have followed the approach of Haverd et al. (2009) who found that the normalized Lagrangian time scale $[(T_L*u_*)/h_c]$ can be parameterized as a function of a normalized length scale within and above the canopy ($z/h_c$) with the shape of an exponential decay function with a constant value: ( $T_L*u_*)/h_c = c_2*(1-\exp(-c_1* (z/h_c)))/(1-\exp(-c_1))$ with $C_1=4.86$; $C_2=0.66$. The Lagrangian time scale is thus calculated as:
$T_L = c_2*(1-\exp(-c_1* (z/h_c)))/(1-\exp(-c_1)) *(h_c/u_*)$. Hence equations 3 and 6 are not in conflict with each other.
We will correct the reference and address this issue in the revised manuscript by improving the description and adding this equation.

[Figure]

**#3 Line 13, page 6: Deff should be CDeff**
Thanks for pointing this out. We will correct this typo in the revised manuscript.

**#  Explain how ksurf is used in the model.**
We have explained the use of $K_{surf}$ in the reply to referee #1 (comment #1) and annotated **Fig.** 1 by Ryder et al. 2016 to illustrate which parameter we are referring to. We will rephrase and add our reply to the manuscript where we discuss equation 7. The more formal description of this parameter is given in the supplementary material of Ryder et al. (2016) in equations S4.30 and S4.31.

**#5 Figures 3 and 4 are nice summaries of overall model performance, but it is unclear how the Taylor scores relate to the magnitude of biases. Sensible heat flux and latent heat flux have low Taylor scores at particular times of the year or times of the day. It would be helpful to have plots of model and observed fluxes for both the annual cycle and the diurnal cycle so that the reader can clearly see the magnitude of the flux biases**
This issue has also been highlighted by referee #1 (comment #6). We will prepare additional figures to show the absolute values of both the simulation and observation at the diurnal and inter-annual scale.

[revised manuscript text omitted]

$u_*$ friction velocity $u_i$ velocity at level $i$ $V_{cmax}$ carboxylation capacity

**Table 1.** Continuation of Table 1

[revised manuscript text omitted]

---

## Author Response (AR2)

Dear Editor,

We would like to thank you for the effort in reviewing this study and the opportunity to prepare a revised manuscript. In line with the comment made by the reviewer#2, we made a modification regarding the symbols on both Fig.5 and Fig.6. The simulation results are filled with colors and the open symbols represent the results from observation. Figure captions of Fig.5 and Fig.6 were also been rewritten and rephrased.

I look forward to hearing from you at your earliest convenience.

Kind Regards,

Yi-Ying Chen on behalf of the author team

Postdoctoral Research Fellow

[revised manuscript text omitted]

---

## Author Response (AR3)

Dear Editor,

We would like to thank you for your suggestions. We moved the svn reversion number to the code availability section and added the version number to the ORCHIDEE-CAN branch. The caption of Fig.5 was also corrected.

I look forward to hearing from you at your earliest convenience.

Kind Regards,

Yi-Ying Chen on behalf of the author team

Postdoctoral Research Fellow

[revised manuscript text omitted]